# A Study of Bayesian Neural Network Surrogates for Bayesian Optimization

**Yucen Lily Li, Tim G. J. Rudner, Andrew Gordon Wilson**
New York University

## Abstract

Bayesian optimization is a highly efficient approach to optimizing objective functions which are expensive to query. These objectives are typically represented by Gaussian process (GP) surrogate models which are easy to optimize and support exact inference. While standard GP surrogates have been well-established in Bayesian optimization, Bayesian neural networks (BNNs) have recently become practical function approximators, with many benefits over standard GPs such as the ability to naturally handle non-stationarity and learn representations for high-dimensional data. In this paper, we study BNNs as alternatives to standard GP surrogates for optimization. We consider a variety of approximate inference procedures for finite-width BNNs, including high-quality Hamiltonian Monte Carlo, low-cost stochastic MCMC, and heuristics such as deep ensembles. We also consider infinite-width BNNs, linearized Laplace approximations, and partially stochastic models such as deep kernel learning. We evaluate this collection of surrogate models on diverse problems with varying dimensionality, number of objectives, non-stationarity, and discrete and continuous inputs. We find: (i) the ranking of methods is highly problem dependent, suggesting the need for tailored inductive biases; (ii) HMC is the most successful approximate inference procedure for fully stochastic BNNs; (iii) full stochasticity may be unnecessary as deep kernel learning is relatively competitive; (iv) deep ensembles perform relatively poorly; (v) infinite-width BNNs are particularly promising, especially in high dimensions.

## 1 Introduction

*Bayesian optimization* (O'Hagan, 1978) is a distinctly compelling success story of Bayesian inference. In Bayesian optimization, we place a prior over the objective we wish to optimize, and use a *surrogate model* to infer a posterior predictive distribution over the values of the objective at all feasible points in space. We then combine this predictive distribution with an *acquisition function* that trades-off exploration (moving to regions of high uncertainty) and exploitation (moving to regions with a high expected value, for maximization). The resulting approach converges quickly to a global optimum, with strong performance in many expensive black-box settings ranging from experimental design, to learning parameters for simulators, to hyperparameter tuning (Frazier, 2018; Garnett, 2023).

While many acquisition functions have been proposed for Bayesian optimization (e.g. Frazier et al., 2008; Wang and Jegelka, 2017), Gaussian processes (GPs) with standard Matérn or RBF kernels are almost exclusively used as the surrogate model for the objective, without checking whether other alternatives would be more appropriate, despite the fundamental role that the surrogate model plays in Bayesian optimization.

Thus, despite promising advances in Bayesian optimization research, there is an elephant in the room: *should we be considering other surrogate models?* It has become particularly timely to evaluate Bayesian neural network (BNN) surrogates as alternatives to Gaussian processes with standard kernels: In recent years, there has been extraordinary progress in making BNNs practical (e.g. Daxberger et al., 2021; Khan and Rue, 2021; Rudner et al., 2022; Tran et al., 2022; Wilson and Izmailov, 2020). Moreover, BNNs can flexibly represent the non-stationary behavior typical of optimization objectives, discover similarity measures as part of representation learning which is useful for higher dimensional inputs, and naturally handle multi-output objectives. In parallel, *Monte-Carlo* acquisition functions (Balandat et al., 2020) have been developed which only require posterior samples, significantly lowering the barrier to using non-GP surrogates that do not provide closed-form predictive distributions.

In this paper, we exhaustively evaluate Bayesian neural networks as surrogate models for Bayesian optimization. We consider conventional fully stochastic multilayer BNNs with a variety of approximate inference procedures, ranging from high-quality full-batch Hamiltonian Monte Carlo (Izmailov et al., 2021; Neal, 1996; 2010), to stochastic gradient Markov Chain Monte Carlo (Chen et al., 2014), to heuristics such as deep ensembles (Lakshminarayanan et al., 2017). We also consider infinite-width BNNs (Lee et al., 2017; Neal and Neal, 1996), corresponding to GPs with fixed non-stationary kernels derived from a neural network architecture, as well as partially Bayesian last-layer deep kernel learning methods (Wilson et al., 2016). This particularly wide range of neural network-based surrogates allows us to evaluate the role of representation learning, non-stationarity, and stochasticity in modeling Bayesian optimization objectives. Moreover, given that so much is unknown about the role of the surrogate model, we believe it is particularly valuable not to have a "horse in the race", such as a special BNN model particularly designed for Bayesian optimization, in order to conduct an unbiased scientific study where any outcome is highly informative.

We also extensively study a variety of synthetic and real-world objectives—with a wide range of input space dimensionalities, single- and multi-dimensional output spaces, and both discrete and continuous inputs, and non-stationarities.

Our study provides several key findings: (1) while stochasticity is often prized in Bayesian optimization (Garnett, 2023; Snoek et al., 2012), due to the small data sizes in Bayesian optimization, fully stochastic BNNs do not consistently dominate deep kernel learning, which is not stochastic about network parameters; (2) of the fully stochastic BNNs, HMC generally works the best for Bayesian optimization, and deep ensembles work surprisingly poorly, given their success in other settings; (3) on standard benchmarks, standard GPs are relatively competitive, due to their strong priors and simple exact inference procedures; (4) there is no single method that dominates across most problems, demonstrating that there is significant variability across Bayesian optimization objectives, where tailoring the surrogate to the objective has particular value; (5) infinite-width BNNs are surprisingly effective at high-dimensional optimization. These results suggest that the non-Euclidean similarity metrics constructed from neural networks are valuable for high-dimensional Bayesian optimization, but representation learning (provided by DKL and finite-width BNNs) is not as valuable as a strong prior derived from a neural network architecture (provided by the infinite-width BNN).

This study also serves as an evaluation framework for considering alternative surrogate models for Bayesian optimization. Our code is available at `https://github.com/yucenli/bnn-bo`.

## 2 RELATED WORK

There is a large body of literature on improving the performance of Bayesian optimization. However, an overwhelming majority of this research only considers Gaussian process surrogate models, focusing on developing new acquisition functions (e.g. Frazier et al., 2008; Wang and Jegelka, 2017), additive covariance functions (Gardner et al., 2017; Kandasamy et al., 2015), using gradient information (Wu et al., 2017), multi-objectives (Swersky et al., 2013), trust region methods that use input partitioning for higher dimensional and non-stationary data (Eriksson et al., 2019), and covariance functions for discrete inputs and strings (Moss et al., 2020). For a comprehensive review, see Garnett (2023).

There has been some prior work focusing on other types of surrogate models for Bayesian optimization, such as random forests (Hutter et al., 2011) and tree-structured Parzen estimators (Bergstra et al., 2013). Snoek et al. (2015) apply a Bayesian linear regression model to the last layer of a deterministic neural network, which can be helpful for the added number of objective queries associated with higher dimensional inputs. Deep kernel learning (Wilson et al., 2016), which transforms the inputs of a Gaussian process kernel with a deterministic neural network, may also be used with Bayesian optimization, especially in specialized applications like protein engineering (Stanton et al., 2022). The linearized-Laplace approximation to produce a linear model from a neural network has also recently been applied to Bayesian optimization (Kristiadi et al., 2023). Neural networks have also been used for Bayesian optimization in the the contextual bandit setting, using the neural tangent kernel for exploration (Zhou et al., 2020; Dai et al., 2022; Lisicki et al., 2022).

Despite the recent practical advances in developing Bayesian neural networks for many tasks (e.g. Wilson and Izmailov, 2020), and recent Monte-Carlo acquisition functions which make it easier to use surrogates like BNNs that do not provide closed-form predictive distributions (Balandat et al., 2020), there is a vanishingly small body of work that considers BNNs as surrogates for Bayesian optimization. This is surprising, since we would indeed expect BNNs to have properties naturally

aligned with Bayesian optimization, such as the ability to learn non-stationary functions without explicit modeling interventions and gracefully handle high-dimensional input and output spaces.

The possible first attempt to use a Bayesian neural network surrogate for Bayesian optimization (Springenberg et al., 2016) came before most of these advances in BNN research, and used a form of stochastic gradient Hamiltonian Monte Carlo (SGHMC) (Chen et al., 2014) for inference. Like Snoek et al. (2015), the focus was largely on scalability advantages over Gaussian processes; however, the reported performance gains were marginal, and puzzling in that they were largest for a *small* number of objective function queries (where the neural net would not be able to learn a rich representation). Kim et al. (2021) used the same method for BNNs with Bayesian optimization, also with SGHMC, targeted at scientific problems with known structures and high dimensionality. In these applications, BNNs leverage auxiliary information, domain knowledge, and intermediate data, which would not typically be available in many Bayesian optimization problems. Foldager et al. (2023) also studied BNN surrogates through mean-field BNNs and deep ensembles, and Müller et al. (2023) used neural network surrogates which approximate the posterior through in-context learning. However, the innate differences between approximate inference methods for BNNs have not been explored.

Our paper provides several key contributions in the context of this prior work, where standard GP surrogates are nearly always used with Bayesian optimization. While finite-width BNN surrogates have been attempted, they are often applied in specialized settings without an effort to understand their properties. Little is known about whether BNNs could generally be used as an alternative to GPs for Bayesian optimization, especially in light of more recent general advances in BNN research. This is the first paper to provide a comprehensive study of BNN surrogates, considering a range of model types, experimental settings, and types of approximate inference. We test the utility of BNNs in a variety of contexts, exploring their behavior as we change the dimensionality of the problem and the number of objectives, investigating their performance on non-stationary functions, and also incorporating problems with a mix of discrete and continuous input parameters. Moreover, we are the first to study infinite BNN models in Bayesian optimization, and to consider the role of stochasticity and representation learning in neural network based Bayesian optimization surrogates. Finally, rather than champion a specific approach, we provide an objective assessment, also highlighting the benefits of GP surrogates for general Bayesian optimization problems.

## 3 SURROGATE MODELS

We consider a wide variety of surrogate models, separately understanding the role of stochasticity, representation learning, and strong priors in Bayesian optimization surrogates. We provide additional information about these surrogates and background about Bayesian optimization in Appendix A.

**Gaussian Processes.** Throughout our experiments, when we refer to Gaussian processes, we always mean *standard* Gaussian processes, with the Matérn-5/2 kernel that is typically used in Bayesian optimization (Snoek et al., 2012). These Gaussian processes have the advantage of simple exact inference procedures, strong priors, and few hyperparameters, such as length-scale, which controls rate of variability. On the other hand, these models are *stationary*, meaning the covariance function is translation invariant and models the objective as having similar properties (such as rate of variation) at different points in input space. They also provide a similarity metric for data points based on simple Euclidean distance of inputs, which is often not suitable for higher dimensional input spaces.

**Fully Stochastic Finite-Width Bayesian Neural Networks.** These models treat all of the parameters of the neural network as random variables, which necessitates approximate inference. We consider three different mechanisms of approximate inference: (1) Hamiltonian Monte Carlo (HMC), an MCMC procedure which is the computationally expensive gold standard (Izmailov et al., 2021; Neal, 1996; 2010); (2) Stochastic Gradient Hamiltonian Monte Carlo (SGHMC), a scalable MCMC approach that works with mini-batches (Chen et al., 2014); (3) Deep Ensembles, considered a practically effective heuristic that ensembles a neural network retrained multiple times, and has been shown to approximate fully Bayesian inference (Izmailov et al., 2021; Lakshminarayanan et al., 2017). These approaches are fully stochastic, and can do *representation learning*, meaning that they can learn appropriate distance metrics for the data as well as particular types of non-stationarity.

**Deep Kernel Learning.** Deep kernel learning (DKL) (Wilson et al., 2016) is a hybrid Bayesian deep learning model, which layers a GP on top of a neural network feature extractor. This approach can do non-Euclidean representation learning, handle non-stationarity, and also uses exact inference. However, it is only stochastic about the last layer.

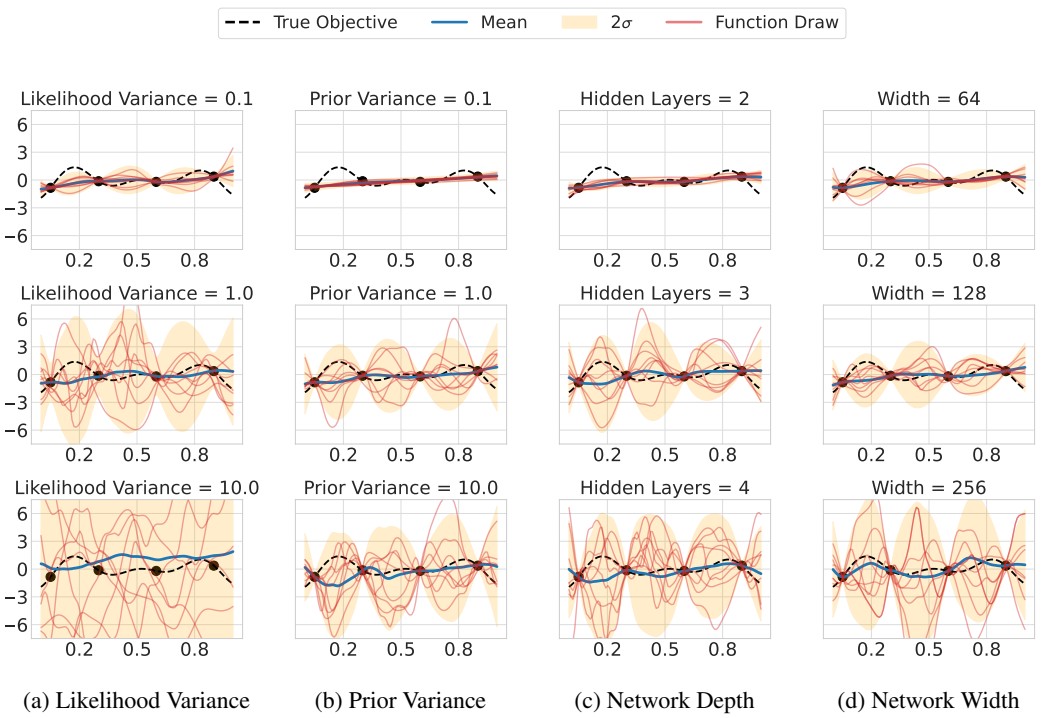

Figure 1: **The design of the BNN has a significant impact on the uncertainty estimates.** We visualize the uncertainty estimates and function draws produced by full-batch HMC on a simple toy objective function with four function queries (denoted in black). For the visualizations above, we fix all other design choices with the following base parameters: likelihood variance = 1, prior variance = 1, number of hidden layers = 3, and width = 128. We see that varying the different aspects of the model leads to significantly different posterior predictive distributions.

**Linearized Laplace Approximation.** The linearized-Laplace approximation (LLA) is a deterministic approximate inference method that uses the Laplace approximation (MacKay, 1992; Immer et al., 2021) to produce a linear model from a neural network, and has recently been considered for Bayesian optimization in concurrent work (Kristiadi et al., 2023).

**Infinite-Width Bayesian Neural Networks.** Infinite-width neural networks (I-BNNs) refer to the behavior of neural networks as the number of nodes per hidden layer increases to infinity. Neal (1996) famously showed with a central limit theorem argument that a BNN with a single infinite-width hidden layer converges to a GP with a neural network covariance function, and this result has been extended to deep neural networks by Lee et al. (2017). I-BNNs are fully stochastic and very different from standard GPs, as they can handle non-stationarity and provide a non-Euclidean notion of distance inspired by a neural network. However, it cannot do representation learning and instead has a fixed covariance function that provides a relatively strong prior.

### 3.1 ROLE OF ARCHITECTURE

We conduct a sensitivity study into the role of key design choices for BNN surrogates. We highlight results for HMC, as it is the gold standard for approximate inference in BNNs (Izmailov et al., 2021).

Gaussian processes involve relatively few design choices—essentially only the covariance function, which is often set to the RBF or Matérn kernel, and we are also able to have an intuitive understanding of what the induced distributions over functions look. In contrast, with BNNs, we must consider the architecture, the prior over parameters, and the approximate inference procedure. It is also less clear how different modeling choices in BNNs affect the inferred posterior predictive distributions. To illustrate these differences for varying network, prior, and variance parameters, we plot the inferred posterior predictive distributions over functions for different network widths and depths, the activation functions, and likelihood and variance parameters in Figure 1, and we evaluate the performance under different model choices for three synthetic data problems in Figure 2. We focus on fully-connected multi-layer perceptrons for this study: while certain architectures have powerful inductive biases

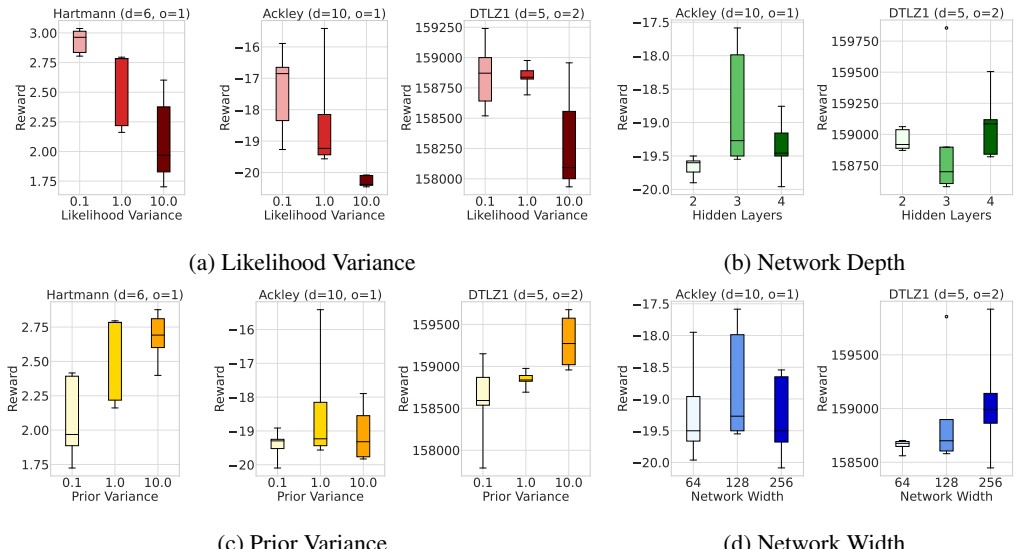

(a) Likelihood Variance      (b) Network Depth

(c) Prior Variance      (d) Network Width

Figure 2: **There is no single architecture for HMC that performs the best across all problems.** We compare the impact of the design on the Bayesian optimization performance for different benchmark problems. For each set of experiments, we fix all other aspects of the design and plot the values of the maximum reward found using HMC after 100 function evaluations over 10 trials.

for vision and language tasks, generic regression tasks such as Bayesian optimization tend to be well-suited for fully-connected multi-layer perceptrons, which have relatively mild inductive biases and make loose assumptions about the structure of the function.

**Model Hyperparameters.** We consider isotropic priors over the neural network parameters with zero mean and variance parameters 0.1, 1, and 10. Similarly, we consider Gaussian likelihood functions with variance parameters 0.1, 1, and 10. The corresponding posterior predictive distributions for full-batch HMC are shown in Figure 1a. As would be expected, an increase in the likelihood variance results in a poor fit of the data and virtually no posterior collapse. In contrast, increasing the prior variance results in a higher predictive variance between data points with a good fit to the data points, whereas a prior variance that is too small leads to over-regularization and uncertainty collapse. As shown in Figure 2a and Figure 2c, lower likelihood variance parameters and larger prior variance parameters tended to perform best across three synthetic data experiments.

**Network Width and Depth.** To better understand the effects of the network size on inference, we explore the performance when varying the number of hidden layers and the number of parameters per layer, each corresponding to an increase in model complexity. In Figure 1c, we see that there is a significant increase in uncertainty as we increase the number of hidden layers. Figure 1d also shows an increase in uncertainty as we increase the width of the network, where a smaller width leads to function draws that are much flatter than function draws from a larger width. However, the best size to choose seems to be problem-dependent, as shown in Figure 2b and Figure 2d.

**Activation Function.** The choice of activation function in a neural network determines important characteristics of the function class, such as smoothness or periodicity. The impact of the activation function can be seen in Appendix D.1, with function draws from the ReLU BNN appearing more jagged and function draws from the tanh BNN more closely resembling the draws from a GP with a Squared Exponential or Matérn 5/2 covariance function.

## 4   EMPIRICAL EVALUATION

We provide an extensive empirical evaluation of BNN surrogates for Bayesian optimization. We first assess how BNNs compare to GPs in relatively simple and well-understood settings through commonly used synthetic objective functions, and we perform an empirical comparison between GPs and different types of BNNs (HMC, SGHMC, LLA, ENSEMBLE, I-BNN, and DKL). To further ascertain whether BNNs may be a suitable alternative to GPs in real-world Bayesian optimization problems, we study six real-world datasets used in prior work on Bayesian optimization with GP

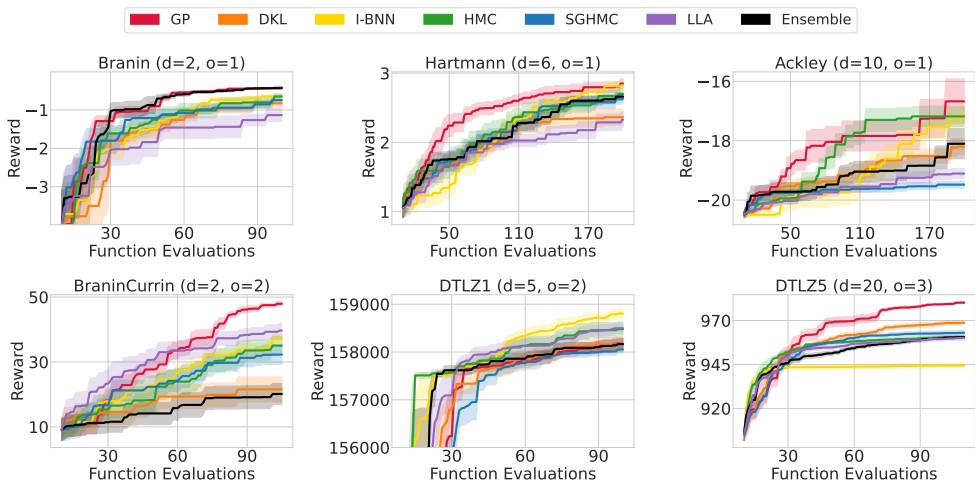

Figure 3: **BNNs are often comparable to GPs on standard synthetic benchmarks.** However, the type of BNN used has a big impact: HMC typically outperforms other BNN approximation methods, while SGHMC and deep ensembles seem to have less reliable performance and are often unable to effectively find the maximum. LLA also has poor performance across the single-objective problems. For each benchmark function, we include $d$ for the number of input dimensions, and $o$ for the number of objectives. We plot the mean and one standard error of the mean over 10 trials.

surrogates (Dreifuerst et al., 2021; Eriksson et al., 2019; Maddox et al., 2021; Oh et al., 2019; Wang et al., 2020). We also provide evidence that the performance of BNNs could be further improved with a careful selection of network hyperparameters. We conclude our evaluation with a case study of Bayesian optimization tasks where simple Gaussian process models may fail but BNN models would be expected to prevail. To this end, we design a set of experiments to assess the performance of GPs and BNNs as a function of the input dimensionality and in settings where the objective function is non-stationary.

## 4.1 SYNTHETIC BENCHMARKS

We evaluate BNN and GP surrogates on a variety of synthetic benchmarks, and we choose problems with a wide span of input dimensions to understand how the performance differs as we increase the dimensionality of the data. We also select problems that vary in the number of objectives to compare the performance of the different surrogate models. Detailed problem descriptions can be found in Appendix B.1, and we include the experiment setup in Appendix C. We use Monte-Carlo based Expected Improvement (Balandat et al., 2020) as our acquisition function for all problems.

As shown in Figure 3, we find BNN surrogate models to show promising results; however, the specific performance of different BNNs varies considerably per problem. DKL matches GPs in Branin and BraninCurrin, but seems to perform poorly on highly non-stationary problems such as Ackley. I-BNNs also seem to slightly underperform compared to GPs on these synthetic problems, many of which have a small number of input dimensions. In contrast, we find finite-width BNNs using full HMC to be comparable to GPs, performing similarly in many of the experiments, slightly underperforming in Hartmann and DTLZ5, and outperforming GPs in the 10-dimensional Ackley experiment. However, this behavior is not generalizable to all approximate inference methods: the performance of SGHMC and LLA vary significantly per problem, matching the performance of HMC and GPs in some experiments while failing to approach the maximum value in others. Deep ensembles also consistently underperform the other surrogate models, plateauing at noticeably lower objective values on problems like BraninCurrin and DTLZ1. This result is surprising, since ensembles are often seen as an effective way to measure uncertainty (Appendix D.3). We provide additional experiments studying how the performance of surrogates changes as we vary their hyperparameters in Appendix D, and we find that these hyperparameters generally have minimal effects on the performance.

## 4.2 REAL-WORLD BENCHMARKS

To provide an evaluation of BNN surrogates in more realistic optimization problems, we consider a diverse selection of real-world applications which span a variety of domains, such as solving

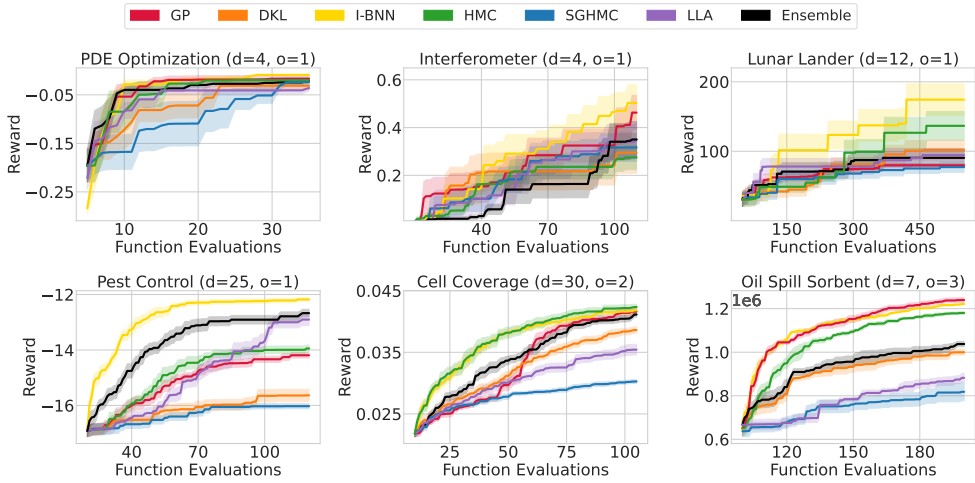

Figure 4: **Real world benchmarks show mixed results**. BNNs outperform GPs on some problems and underperform on others, and there does not seem to be a noticeable preference for any particular surrogate as we increase the number of input dimensions. Additionally, there does not appear to be a clear separation between the top row of experiments, which optimize over continuous parameters, and the bottom row of experiments, which also include some discrete inputs. For each benchmark, we include $d$ for the number of input dimensions, and $o$ for the number of objectives. We plot the mean and one standard error of the mean over 10 trials.

differential equations and monitoring cellular network coverage (Dreifuerst et al., 2021; Eriksson et al., 2019; Maddox et al., 2021; Oh et al., 2019; Wang et al., 2020). Many of these problems, such as the development of materials to clean oil spills, have consequential applications; however, these objectives are often multi-modal and are difficult to optimize globally. Additionally, unlike the synthetic benchmarks, many real-world applications consist of input data with ordinal or categorical values, which may be difficult for GPs to handle. Several of the problems also require multiple objectives to be optimized. Detailed problem descriptions are provided in Appendix B.2.

We share the results of our experiments in Figure 4, and details about the experiment setup can be found in Appendix C. The results are mixed: BNNs are able to significantly outperform GPs in the Pest Control dataset, while GPs find the maximum reward in the Cell Coverage and Lunar Lander experiments. The Pest Control, Cell Coverage, and Oil Spill Sorbent experiments all include discrete input parameters, and there seems to be a slight trend of GP and I-BNNs performing well, and SGHMC and LLA performing more poorly. Similar to the findings from the synthetic benchmarks, we see that the different approximate inference methods for finite-width BNNs lead to significantly different Bayesian optimization performance, with HMC generally finding higher rewards compared to SGHMC, LLA, and deep ensembles. Additionally, it appears that GPs perform well in the two multi-objective problems, although that may not be generalizable to additional multi-objective problems and may be more related to the curvature of the specific problem space.

## 4.3 LIMITATIONS OF GP SURROGATE MODELS

Although popular, GPs suffer from well-known limitations that directly impact their usefulness as surrogate models. To contrast BNN and GP surrogates, we explore two failure modes of GPs and demonstrate that BNN surrogate models can overcome these issues.

**Non-Stationary Objective Functions.** To use GPs, we must specify a kernel function class that governs the covariance structure over data points. We typically constrain models to have kernels of the form $k(\mathbf{x}, \mathbf{x}') = k(\|\mathbf{x} - \mathbf{x}'\|)$ because it is easier to describe the functional form and learn the hyperparameters of the kernel. However, because the covariance between two values only depends on their distance and not on the values themselves, this setup assumes the function is stationary and has similar mean and smoothness throughout the input space. Unfortunately, this assumption does not hold true in many real-world settings. For example, in the common Bayesian optimization application of choosing hyperparameters of a neural network, the true loss function landscape may have vastly different behavior in one part of the input space compared to another. BNN surrogates, in contrast to GP surrogates, are able to model non-stationary functions without similar constraints.

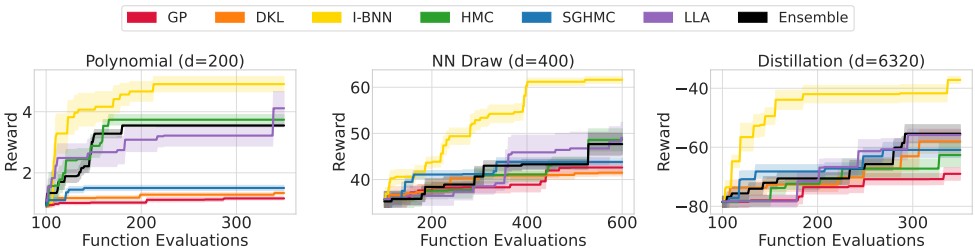

Figure 5: **I-BNNS outperform other surrogates in many high-dimensional settings**. We show the results of maximizing a polynomial function (left), maximizing a fixed function draw from a neural network (center), and optimizing the parameters of a neural network in the context of knowledge distillation (right). All of these objectives are high-dimensional and non-stationary, and we find that BNNs consistently find higher rewards than GPs across all problems. We plot the mean and one standard error of the mean over 10 trials, and $d$ corresponds to the number of input dimensions.

In Appendix D.9, we show the performance of BNN and GP surrogate models for a non-stationary objective function. Because the GP assumes that the behavior of the function is the same throughout the input domain, it cannot accurately model the input-dependent variation and underfits around the true optimum. In contrast, BNN surrogates can learn the non-stationarity of the function.

**High-Dimensional Input Spaces.** Due to the curse of dimensionality, GPs do not scale well to high-dimensional input spaces without careful human intervention. Common covariance functions may fail to faithfully represent high-dimensional input data, making the design of custom-tailored kernel functions necessary. In contrast, neural networks are well-suited for modeling high-dimensional input data (Krizhevsky et al., 2012).

To measure the effect of dimensionality on the performance of GPs and BNNs, we use synthetic test functions provided by high-dimensional polynomial functions and function draws from neural networks. We also construct a realistic high-dimensional problem by using Bayesian optimization to set the parameters of a neural network in the context of knowledge distillation. Knowledge distillation refers to the act of "distilling" information from a larger teacher model to a smaller student model by matching model outputs (Hinton et al., 2015), and it is known to be a difficult optimization problem (Stanton et al., 2021). For full descriptions, see Appendix B.

We share the results of our findings in Figure 5 and Appendix D.4. We see I-BNNs clearly stand out in these high-dimensional settings. The I-BNN has several appealing features in this setting: (1) it provides a non-Euclidean and non-stationary similarity metric, which can be particularly valuable in high-dimensions; (2) it does not have any hyperparameters for learning, and thus is not "data hungry"—which is especially important in high dimensional problems with small data sizes, since these settings provide relatively little information for representation learning. Additionally, we find that other BNN surrogate models also outperform GPs across the high-dimensional problems, providing a compelling motivation for BNNs as surrogate models for Bayesian optimization.

## 4.4 Model Rankings

To further interpret our findings, we visualize the model performance in Figure 6. We determine the relative performance of each model by $r_i$. For each trial, let $r_h$ denote the highest maximum reward found by any model and $r_l$ denote the lowest. The score of model with maximum reward is $(r_i - r_h)/(r_h - r_l)$.

We plot the distribution of scores in Figure 6. Across all experiments, I-BNNs and GPs have competitive performance, while SGHMC and DKL perform more poorly. However, in high-dimensional settings, I-BNNs outperform all other surrogate models. GPs, in contrast, perform poorly in this setting, consistently finding lower rewards than BNN surrogates.

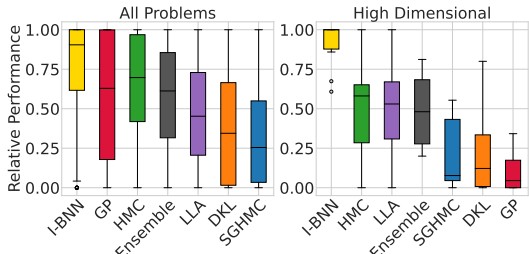

Figure 6: **We rank models by their relative performance.** Across all problems, GPs and I-BNNs have strong relative performance. However, in high-dimensional problems, I-BNNs consistently outperform other surrogate models while GPs tend to find much lower rewards.

## 4.5 ADDITIONAL PRACTICAL CONSIDERATIONS

**Network Architecture.** To further understand the impact of network architecture on the performance of BNNs, we conduct an extensive neural architecture search over a selection of the benchmark problems. While a thorough search is often impractical for Bayesian optimization problems, we use this experiment to investigate the flexibility of BNNs. We find that the performance of BNNs can significantly increase for problems such as Pest Control, demonstrating the usefulness of BNN for Bayesian optimization when the architecture is well-suited for a given problem. We provide additional details and showcase the performance of BNNs with different architectures in Appendix D.1.

**Quality of Mean and Uncertainty Estimates.** To better understand the quality of the mean and uncertainty estimates of different surrogate models, we conducted ablation studies for which we created hybrid models that combine the mean estimate of one surrogate with the uncertainty estimate of another. By comparing the performance of a given surrogate model to hybrid models that have the same mean as the non-hybrid model but the uncertainty estimates of a different surrogate model (or the other way around), we find that HMC and I-BNN surrogates often have better mean estimates compared to GPs while GPs have better uncertainty estimates, which we detail in Appendix D.2.

**Large Number of Function Queries.** We investigate the effect of performing a larger number of function queries on performance when using BNN and GP surrogate models. We find in this setting that (1) I-BNNs remain competitive; (2) BNNs perform well, leveraging the data for representation learning; (3) the performance of deep ensembles is greatly improved, consistent with the explanation that their poor performance on many tasks is due to limited data. We share details in Appendix D.10.

**Runtime.** In real-world Bayesian optimization problems, querying the objective function is typically significantly more time-consuming than (re-)training a surrogate model after new data has been obtained, making the quality of the surrogate model paramount and the time needed for (re-)training the surrogate model of secondary interest. Nevertheless, given the varying training requirements of BNNs and GPs, we provide wall-clock times of all surrogate models across our experiments in Appendix D.11. Notably, I-BNNs are particularly competitive both in performance and runtime.

## 4.6 REVISITING STANDARD ASSUMPTIONS

While Gaussian processes are typically used as the default surrogate model, there are many design choices, such the choice of kernel and the method of hyperparameter selection, which play a crucial role. It is commonly prescribed to use the Matérn kernel and perform hyperparameter marginalization rather than optimization (e.g. Eriksson et al., 2019; Snoek et al., 2012). In Figure A.13 and Figure A.12, we compare the performance of different specifications of GPs across our diverse benchmarks. In contrast to conventional wisdom, we do not find that using the Matérn kernel and hyperparameter marginalization significantly improves the performance of GPs in general; in fact, there are many problems where the RBF kernel or hyperparameter optimization are preferable.

## 5 DISCUSSION

While Bayesian optimization research has made significant progress over the last few decades (Garnett, 2023), the surrogate model remains a crucial yet underexplored design choice, with standard GPs being viewed as default surrogates. Given recent advances in BNNs and related approaches, it is, therefore, particularly timely to consider neural network surrogates for Bayesian optimization.

Although we found that BNNs are competitive with standard GPs for Bayesian optimization, our findings that DKL is competitive with BNNs, and that infinite-width BNNs show promising performance in general—but especially for higher dimensional settings—raise the question of whether a fully stochastic treatment of finite BNN surrogates is typically necessary for Bayesian optimization. Since infinite-width BNNs do not involve learning many hyperparameters and do not require approximate inference, they are well-positioned to become a de facto standard surrogate for Bayesian optimization.

Finally, we found that, on the one hand, no single surrogate model consistently outperforms all other surrogate models across all Bayesian optimization problems considered in our evaluation. This finding supports the use of simple models with strong but generic assumptions—such as standard GP models. On the other hand, the standard benchmarking problems were designed with only GP surrogates in mind, and we found that standard GPs underperform BNNs and other neural network-based surrogates on challenging high-dimensional problems.

**Acknowledgements.** We thank Greg Benton for helpful guidance in the beginning stages of this research, and Sanyam Kapoor for discussions. This work is supported by NSF CAREER IIS-2145492, NSF I-DISRE 193471, NIH R01DA048764-01A1, NSF IIS-1910266, NSF 1922658 NRT-HDR, Meta Core Data Science, Google AI Research, BigHat Biosciences, Capital One, and an Amazon Research Award.

## 6 REPRODUCIBILITY

We provide the code needed to reproduce all experiments in the supplementary material attached, and we also provide experiment details for all of our results in Appendix C. We also release our code on GitHub: `https://github.com/yucenli/bnn-bo`.

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

# Appendix

## A Study of Bayesian Neural Network Surrogates
## for Bayesian Optimization

### TABLE OF CONTENTS

## APPENDIX A    BACKGROUND

### A.1    BAYESIAN OPTIMIZATION

Bayesian optimization is a global optimization method commonly used for black-box functions which are costly to evaluate. Specifically, we formulate the optimization as a maximization problem where we want to solve $\mathbf{x}^* = \arg\max_{\mathbf{x}} f(\mathbf{x})$, where $\mathbf{x}$ represents all possible inputs to objective function $f$, using a limited number of function queries. In Bayesian optimization, we iteratively select new function evaluations using the following procedure: (1) use a *surrogate model* to find $p(f|\mathcal{D})$, where $\mathcal{D} = \{\mathbf{x}_i, y_i\}_{i=1}^t$ and $y_i$ is a noisy observation of $f(\mathbf{x}_i)$; (2) select the next $\mathbf{x}_{t+1}$ to query by maximizing an *acquisition function*, which is an inexpensive computation that typically uses the mean and variance of the posterior $p(f|\mathcal{D})$ to compute how useful a function evaluation would be; and (3) query the function at value $\mathbf{x}_{t+1}$ to observe $y_{t+1} \sim \mathcal{N}(f(\mathbf{x}_{t+1}), \sigma_{\text{obs}}^2)$ and add the result to the dataset $\mathcal{D} := \mathcal{D} \cup \{\mathbf{x}_{t+1}, y_{t+1}\}$.

Historically, Bayesian optimization has seen wide success in a range of fields including drug design, engineering challenges, and hyperparameter optimization. A more recent and growing area of interest is in higher dimensional settings, where the objective, $f(x)$ is multi-dimensional consisting of several tasks we wish to optimize jointly. In multi-objective Bayesian optimization, we want to find the input $\mathbf{x}^*$ that maximizes $k$ related objective functions $\mathcal{F} = \{f_1(\mathbf{x}), \cdots, f_k(\mathbf{x})\}$. There is typically no single solution that maximizes all $K$ objectives simultaneously. Therefore, performance is typically compared using *Pareto dominance*, where one solution Pareto-dominates another if it performs equally well or better on all objectives. Pareto dominance can be measured using *hypervolume*, which indicates how much of a bounded region is dominated by a solution.

Core to the Bayesian optimization procedure is the use of an acquisition function to select the next candidate points given a set of observations $x_i, f(x_i)_{i=1}^N$, and a surrogate model trained to fit these observations. Acquisition functions are functions of the predictive distribution of the surrogate model, and seek candidate points according to criteria such as the probability of improvement or the expected improvement over the currently found minimizer of the objective.

The chosen surrogate model will also have a signficant impact on the performance of Bayesian optimization . Since the true function $f$ may take a variety of forms, the surrogate model should be flexible enough to accurately represent it. Additionally, the uncertainty estimates that the surrogate model provides should be well-calibrated to ensure that the input that maximizes $f$ can be found in a limited number of iterations.

### A.2    GAUSSIAN PROCESSES

Gaussian processes (GPs) (Rasmussen and Williams, 2006) are distributions over functions entirely specified by a mean function $\mu(\mathbf{x})$ and a covariance function $k(\mathbf{x}, \mathbf{x}')$. For regression tasks with GPs, we assume $y(\mathbf{x}) = f(\mathbf{x}) + \epsilon$ with $f(\mathbf{x}) \sim \mathcal{GP}(\mu(\mathbf{x}), k(\mathbf{x}, \mathbf{x}'))$ and $\epsilon \sim \mathcal{N}(0, \sigma^2)$, where $\sigma^2$ is the likelihood variance. The functions $\mu(\mathbf{x})$ and $k(\mathbf{x}, \mathbf{x}')$ are the mean and kernel functions of the GP that govern the functional properties and generalization performance of the model. In practice, we tune the hyperparameters of $\mu(\mathbf{x})$ and $k(\mathbf{x}, \mathbf{x}')$ on the training data to optimize the *marginal likelihood* of the GP, which maximizes the probability that the GP model will have generated the data.

By the definition of a GP, for any finite collection of inputs $\mathbf{x} = [\mathbf{x}_{\text{train}}, \mathbf{x}_{\text{test}}]$, $f(\mathbf{x})$, and thus $y(\mathbf{x})$ are jointly normal. Therefore, we can apply the rules of conditioning partitioned multivariate normals to form a posterior distribution $p\left(f(\mathbf{x}_{\text{test}}) \mid y(\mathbf{x}_{\text{train}})\right)$, which will also be Gaussian.

In the context of Bayesian optimization, this simple conditioning procedure means that given some set of points at which we have already queried the objective function, we can quickly generate a posterior over potential candidate points and, with the help of an acquisition function, select the next points to query the objective.

### A.3    BAYESIAN NEURAL NETWORKS

Bayesian neural networks (BNNs) are neural networks with stochastic parameters for which a posterior distribution is inferred using Bayesian inference.

More specifically, for regression tasks, we can specify a Gaussian observation model, $p(y(\mathbf{x}) \mid \mathbf{x}, \boldsymbol{\theta}) = \mathcal{N}(h(\mathbf{x} \mid \boldsymbol{\theta}), \sigma^2)$, where $y(\mathbf{x})$ represents a noisy observed value and $h(\mathbf{x} \mid \boldsymbol{\theta})$ represents the output of a neural network with parameter realization $\boldsymbol{\theta}$ for input $\mathbf{x}$. For BNNs, a prior distribution is placed over the stochastic parameters $\boldsymbol{\theta}$ of the neural network, and the corresponding posterior distribution is given by $p(\boldsymbol{\theta} \mid \mathcal{D}) = p(\mathcal{D} \mid \boldsymbol{\theta})p(\boldsymbol{\theta})/p(\mathcal{D})$. This posterior distribution can then be used in combination

with the acquisition function for Bayesian optimization to select the next set of candidate points to query.

There are many different choices to consider when using Bayesian neural networks, such the inference method used to compute the posterior distribution, the architecture of the neural network, and the selection of which parameters are stochastic. In this work, we study the performance of five different types of BNNs with varying inference methods and stochasticity.

### A.3.1 FULLY-STOCHASTIC FINITE-WIDTH NEURAL NETWORKS

For fully-stochastic Bayesian neural networks, every parameter of the neural network is stochastic and has a prior placed over it. Exact inference over these stochastic network parameters for fully-stochastic finite-width BNNs is analytically intractable, since neural networks are non-linear in their parameters. To enable Bayesian inference in this setting, approximate inference methods can be used to find approximations to the exact posterior $p(\boldsymbol{\theta} \mid \mathcal{D})$. In this work, we focus on four types of approximate inference: Hamiltonian Monte Carlo (HMC; Neal (2010)), stochastic-gradient HMC (Chen et al., 2014), linearized Laplace approximations (Immer et al., 2021), and deep ensembles of deterministic neural networks (Lakshminarayanan et al., 2017).

**Hamiltonian Monte Carlo.**  Hamiltonian Monte Carlo is a Markov Chain Monte Carlo method that produces asymptotically exact samples from the posterior distribution (Neal, 2010) and is commonly referred to as the "gold standard" for inference in BNNs. At test time, HMC approximates the integral $p(y(\mathbf{x}_{\text{test}}) \mid \mathcal{D}) \approx \frac{1}{M} \sum_{i=1}^{M} p(y(\mathbf{x}_{\text{test}}) \mid \boldsymbol{\theta}_i)$ using samples $\boldsymbol{\theta}_i$ drawn from the posterior over parameters.

Full-batch HMC provides the most accurate approximation of the posterior distribution but is computationally expensive and in practice limited to models with only a few hundred thousand parameters (Izmailov et al., 2021). Full-batch HMC allows us to study the performance of BNNs in Bayesian optimization without many of the confounding factors of inaccurate approximations of the predictive distribution.

**Stochastic Gradient Hamiltonian Monte Carlo.**  Stochastic gradient methods (Welling and Teh, 2011; Hoffman et al., 2013) are commonly used as an inexpensive approach to sampling from the posterior distribution. Unlike full-batch HMC, which computes the gradients over the full dataset, stochastic gradient HMC instead samples a mini-batch to compute a noisy estimate of the gradient (Chen et al., 2014). While these methods may seem appealing when working with large models or datasets, they can result in inaccurate approximations and posterior predictive distribution with unfaithful uncertainty representations.

**Deep Ensembles.**  Deep ensembles are ensembles of several deterministic neural networks, each trained using maximum a posteriori estimation using a different random seed and initialization (and sometimes using different subsets of the training data). The ensemble components can be viewed as forming an efficient approximation to the posterior predictive distribution, by choosing parameters that represent typical points in the posterior and have high functional variability (Wilson and Izmailov, 2020). Deep ensembles are easy to implement in practice and have been shown to provide highly accurate predictions and a good predictive uncertainty estimate (Lakshminarayanan et al., 2017; Ovadia et al., 2019).

### A.3.2 FULLY-STOCHASTIC INFINITE-WIDTH NEURAL NETWORKS

Infinitely-wide neural networks (Neal and Neal, 1996) refer to the behavior of neural networks when the number of nodes in each internal layer increases to infinity. Each one of these nodes continues is stochastic and has a specific prior placed over its parameters.

**Infinite-width Neural Network.**  It is possible to specify a GP that has an exact equivalence to an infinitely-wide fully-connected Bayesian neural network (I-BNN) (Lee et al., 2017). For a single-layer network, the Central Limit Theorem can be used to show that in the limit of infinite width, each output of the network will be Gaussian distributed, and the exact distribution can be computed (Neal and Neal, 1996; Williams, 1996). This process can then be applied recursively for additional hidden layers (Lee et al., 2017). I-BNNs can be used in the same way as GPs to calculate the posterior distribution for infinitely-wide neural networks.

### A.3.3 LINEARIZED FINITE-WIDTH NEURAL NETWORKS

**Linearized Laplace Approximation.** The linearized Laplace approximation (Immer et al., 2021) (LLA) is a deterministic approximate inference method which uses the Laplace approximation (MacKay, 1992; Immer et al., 2021) to produce a linear model from a neural network. LLAs can be represented as Gaussian processes with mean functions provided by the MAP predictive function, and covariance functions provided by the finite, empirical neural tangent kernel at the MAP estimate.

### A.3.4 PARTIALLY-STOCHASTIC FINITE-WIDTH NEURAL NETWORKS

For partially-stochastic neural networks, we learn point estimates for a subset of the parameters, and we learn the full posterior distribution for the remaining parameters in the neural network.

**Deep Kernel Learning.** Deep Kernel Learning (DKL) is a hybrid method that combines the flexibility of GPs with the expressivity of neural networks (Wilson et al., 2016). In DKL, a neural network $g_{\mathbf{w}}(\cdot)$ parameterized by weights $\mathbf{w}$ is used to transform inputs into intermediate values, where additional GP kernels can then be applied. Specifically, given inputs $\mathbf{x}$ and a base kernel $k_{\text{BASE}}(\mathbf{x}, \mathbf{x}')$, $k_{\text{DKL}}(\mathbf{x}, \mathbf{x}') = k_{\text{BASE}}(g_{\mathbf{w}}(\mathbf{x}), g_{\mathbf{w}}(\mathbf{x}'))$. Unlike GPs with standard kernels which depend only on the distance between inputs and therefore assume that the mean and smoothness of the function are consistent throughout, DKL does not assume stationarity and is able to represent how the properties of the function change due to its neural network feature extractor.

## APPENDIX B    PROBLEM DESCRIPTIONS

### B.1    SYNTHETIC DATASETS

In the single-objective case, **Branin** is a function with two-dimensional inputs with three global minima, **Hartmann** is a six-dimensional function with six local minima, and **Ackley** is a multi-dimensional function with many local minima which is commonly used to test optimization algorithms. We convert all problems to maximization problems, and the goal of Bayesian optimization in single-objective settings is to find the maximum value of the objective function.

For multi-objective Bayesian optimization benchmarks, **BraninCurrin** is a two-dimensional input and two-objective problem composed of the Branin and Currin functions, and **DTLZ1** and **DTLZ5** are multi-dimensional and multi-objective test functions which are used to measure an optimization algorithm's ability to converge to the Pareto-frontier. Here, the goal is to find an input corresponding to the multi-dimensional objective value with the maximum hypervolume from a problem-specific reference point.

### B.2    REAL-WORLD DATASETS

**Interferometer** In this problem, the goal is to tune an optical interferometer through the alignment of two mirrors. We use the simulator in Sorokin et al. (2020) to replicate the Bayesian optimization problem in Maddox et al. (2021). Each mirror has a continuous x and y coordinate, which should be optimized so that the two mirrors reflect light with minimal amounts of interference.

**Lunar Lander** Following Eriksson et al. (2019), we aim to learn the parameters of a controller for a lunar lander as implemented in OpenAI gym. The controller has 12 continuous input dimensions, corresponding to events such as "change the angle of the rover if it is tilted more than $x$ degrees." The objective is to maximize the average final reward over 50 randomly generated environments.

**Cellular Coverage** We want to optimize the cellular network coverage and capacity from 15 cell towers (Dreifuerst et al., 2021). Each tower has a continuous parameter corresponding to transmit power parameter and an ordinal parameter with 6 values corresponding to tilt, for a total of 15 continuous and 15 ordinal input parameters. There are two different objectives: maximize the cellular coverage, and minimize the total interference between cell towers.

**Oil Spill Sorbent** In this problem, we optimize the properties of a material to maximize its performance as a sorbent for marine oil spills (Wang et al., 2020). We tune 5 ordinal parameters and 2 continuous parameters which control the manufacturing process of electrospun materials, and optimize over three objectives: water contact angle, oil absorption capacity, and mechanical strength.

**Pest Control** Minimizing the spread of pests while minimizing the prevention costs of treatment is an important problem with many parallels (Oh et al., 2019). In this experiment, we define the setting as a categorical optimization problem with 25 categorical variables corresponding to stages of intervention, with 5 different values at each stage. We optimize over two objectives: minimizing the spread of pests and minimizing the cost of prevention.

**PDE Optimization** Here, following Maddox et al. (2021), we optimize 4 continuous parameters corresponding to the diffusivity and reaction rates of a Brusselator with spatial coupling. The objective of the problem is to minimize the variance of the PDE output.

### B.3    HIGH-DIMENSIONAL PROBLEMS

**Polynomial** In this optimization problem, the goal is to find the maximum value of a polynomial function. For input $\mathbf{x} \in \mathbb{R}^d$, the objective value is calculated using $\sum_{i=1}^{d/4} \prod_{j=1}^{4} (\mathbf{x}_{i+j} - c_{i+j})$, where $c_i \sim \text{Normal}(0, 1)$. We set the boundaries of the input space to be $[0, 1]^d$, and this problem setup can be used for any number of dimensions $d$.

**Neural Network Draw** We want to find the maximum value of a function specified by a draw from a neural network. For the neural network, we use an MLP with $d$ inputs connected to 2 hidden layers of 256 nodes each with tanh activation, connected to a final layer of size 1 (unless otherwise specified). We set each parameter $w$ in the network to a value drawn from $\mathcal{N}(0, 1)$. The final objective function is equivalent to the output of the neural network with the specified weights. With this setup, we can vary $d$ to alter the input dimensionality of the problem.

**Knowledge Distillation** The goal of knowledge distillation is to use a larger teacher model to train a smaller student model, typically done by matching the teacher and student predictive distributions. In this problem, we use Bayesian optimization to determine the optimal parameters of the student neural

network by setting our objective function as the KL divergence between the teacher and student predictive distributions. Knowledge distillation is known to be a difficult optimization problem, and this is problem also has many more dimensions than typical Bayesian optimization benchmarks. For our experiment, we use the MNIST dataset, and we train a LeNet-5 for our teacher model. For our student model, we use a small CNN with the following architecture:

1. convolutional layer with 16 convolution kernels of 3x3 (ReLU activation)
2. max pool layer 2x2
3. convolutional layer with 12 convolution kernels of 3x3 (ReLU activation)
4. max pool layer 2x2
5. fully connected layer with 10 outputs

## APPENDIX C    EXPERIMENT DETAILS

### C.1    GENERAL SETUP

For all datasets, we normalize input values to be between [0, 1] and standardize output values to have mean 0 and variance 1. We also use Monte-Carlo based Expected Improvement as our acquisition function.

**GP**: For single-objective problems, we use GPs with a Matérn $5/2$ kernel, adaptive scale, a length-scale prior of $\text{Gamma}(3, 6)$, and an output-scale prior of $\text{Gamma}(2.0, 0.15)$, which combine with the marginal likelihood to form a posterior which we optimize for hyperparameter learning. For multi-objective problems, we use the same GP to independently model each objective.

**I-BNN**: We use I-BNNs with 3 hidden layers and the ReLU activation function. We set the variance of the weights to 10.0, and the variance of the bias to 1.6. For multi-objective problems, we independently model each objective.

For each of the following surrogate models which include neural networks, we use MLPs with 3 hidden layers and 128 nodes each. During each iteration of Bayesian optimization, we use a grid search over prior variance (0.1, 1.0, 10.0) and likelihood variance (0.1, 0.32, 1.0) to find the combination which maximizes $\mathcal{L}$, where $\mathcal{L}$ represents the likelihood of the surrogate model on a random 20% of the existing function evaluations when the model is trained on the other 80%.

**DKL**: We set up the base kernel using the same Matérn $5/2$ kernel that we use for GPs. For the feature extractor, we use the model parameters as explained above. For multi-objective problems, we independently model each objective.

**HMC**: We use HMC with an adaptive step size, and we choose the architecture as explained above. We model multi-objective problems by setting the number of nodes in the final layer equal to the number of objectives.

**SGHMC**: We use SGHMC with minibatch size of 5 and neural network architecture as indicated above. We follow the implementation in Springenberg et al. (2016) and use scale-adaptive SGHMC with a heteroskedastic likelihood variance as determined by the output of the neural network. We model multi-objective problems by setting the number of nodes in the final layer equal to the number of objectives.

**LLA**: We use the model architecture as explained. We model multi-objective problems by setting the number of nodes in the final layer equal to the number of objectives.

**ENSEMBLE**: We use an ensemble of 5 models, each with the architecture explained above. Each model is trained on a random 80% of the function evaluations. We model multi-objective problems by setting the number of nodes in the final layer equal to the number of objectives.

We run multiple trials for all experiments, where each trial starts with a different set of initial function evaluations drawn using a Sobol sampler.

### C.2    SYNTHETIC BENCHMARKS

**Branin** We ran 10 trials using batch size 5 with 10 initial points.

**Hartmann** We ran 10 trials using batch size 10 with 10 initial points.

**Ackley** We ran 10 trials using batch size 10 with 10 initial points.

**BraninCurrin** We ran 10 trials using batch size 10 with 10 initial points.

**DTLZ1** We ran 10 trials using batch size 5 with 10 initial points.

**DTLZ5** We ran 10 trials using batch size 1 with 10 initial points.

### C.3    REAL-WORLD BENCHMARKS

**PDE Optimization** We ran 10 trials using batch size 1 with 5 initial points.

**Interferometer** We ran 10 trials using batch size 10 with 10 initial points.

**Lunar Lander** We ran 10 trials using batch size 50 with 50 initial points.

**Pest Control** We ran 10 trials using batch size 4 with 20 initial points.

**Cell Coverage** We ran 10 trials using batch size 5 with 10 initial points.

**Oil Spill Sorbent** We ran 10 trials using batch size 10 with 10 initial points.

## C.4 HIGH-DIMENSIONAL EXPERIMENTS

**Polynomial** We ran 10 trials using batch size 10 with 100 initial points.

**Neural Network Draw** We ran 10 trials using batch size 10 with 100 initial points.

**Knowledge Distillation** We ran 10 trials using batch size 10 with 100 initial points.

## C.5 NEURAL ARCHITECTURE SEARCH

We used SMAC (Lindauer et al., 2022) to find the optimal hyperparameters of HMC for Cell Coverage, Pest Control, and DTLZ5 benchmark problems using Bayesian optimization. For each benchmark, our new objective function was the average maximum value found for three runs of Bayesian optimization using the same problem setup as detailed above. We used the hyperparameter optimization facade in SMAC, and for each problem, we used Bayesian optimization to select 200 different HMC architectures to find the optimal combination from the following set of possible values:

- Network width: [32, 512] (continuous)
- Network depth: {1, 2, 3, 4, 5, 6} (discrete)
- Network activation: {"ReLU", "tanh"} (discrete)
- $\log_{10}$ of likelihood variance: [-3.0, 2.0] (continuous)
- $\log_{10}$ of prior variance: [-3.0, 2.0] (continuous)

We then use the optimal architecture and run the benchmark for 10 trials with the same setup as described in Appendix C.2 and Appendix C.3 to compare the results of HMC with and without neural architecture search.

For cell coverage, the final HMC model was an MLP with 5 hidden layers, 184 parameters per layer, tanh activation, likelihood variance of 26.3, and prior variance of 0.54.

For pest control, the final HMC model was an MLP with 1 hidden layer of size 321, tanh activation, likelihood variance of 0.26, and prior variance of 0.31.

For DTLZ5, the final HMC model was an MLP with 3 hidden layers, 297 parameters per layer, relu activation, likelihood variance of 0.01, and prior variance of 0.30.

## APPENDIX D    FURTHER EMPIRICAL RESULTS

### D.1    BNN ARCHITECTURE

#### D.1.1    NEURAL ARCHITECTURE SEARCH

To further investigate the impact of architecture on the performance of BNNs, we conduct an extensive neural architecture search over a selection of the benchmark problems, varying the width, depth, prior variance, likelihood variance, and activation function. For our experiments, we use SMAC3 (Lindauer et al., 2022), a framework which uses Bayesian optimization to select the best hyperparameters, and we detail the experiment setup in Appendix C. While a thorough search over architectures is often impractical for realistic settings of Bayesian optimization since it requires a very large number of function evaluations, we use this experiment to demonstrate the flexibility of BNNs and to showcase its potential when the design is well-suited for the problem.

We show the effect of neural architecture search on HMC surrogate models in Figure A.1. On the Cell Coverage problem, the architecture search did not drastically change the performance of HMC. In contrast, extensively optimizing the hyperparameters made a significant difference on the Pest Control problem, leading HMC finding higher rewards than GPs while using fewer function evaluations; however, on this problem, I-BNN, which does not require specifying an architecture, still performs best. Neural architecture search was also able to improve the results on DTLZ5, leading HMC to be competitive with other surrogate models such as I-BNNs and DKL. The difference in the benefits of the search may be attributed to some problems having less inherent structure than others, where extensive hyperparameter optimization may not be as necessary. Additionally, our original HMC surrogate model choice may already have been a suitable choice for some problems, so an extensive search over architectures may not significantly improve the performance.

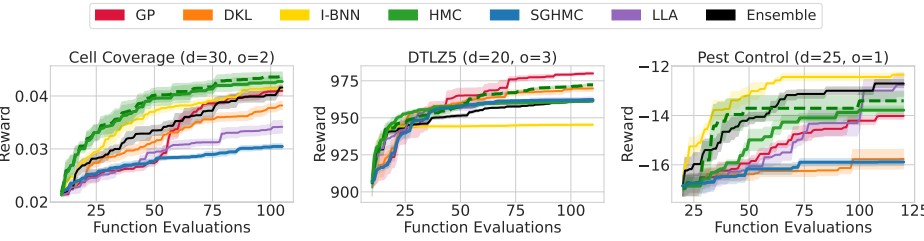

Figure A.1: **The impact of neural architecture search on HMC is problem-dependent.** The dashed green line indicates the performance of HMC after an extensive neural architecture search, compared to the solid green line representing the HMC model selected from a much smaller pool of hyperparameters. We see that it has minimal impact on Cell Coverage (left), moderate impact on DTLZ5 (center), and extensive impact on Pest Control (right), even outperforming GPs. For each benchmark, we include $d$ for the number of input dimensions, and $o$ for the number of objectives. We plot the mean and one standard error of the mean over 10 trials.

### D.1.2 SMALLER ARCHITECTURE

Here, we benchmark the performance of BNNs using the relatively small architecture specified by Kristiadi et al. (2023): an MLP with 2 hidden layers of size 50 with ReLU activation. In the main text, our experiments used MLPs with 3 hidden layers of size 128 with tanh activation. Our experiments show that the larger BNN from our main text typically leads to better performance across datasets. However, even with the smaller architecture, we find that the relative performance of the surrogate models mostly remains consistent. We still find that HMC often outperforms other approximate inference methods such as deep ensembles, and we see that SGHMC often finds suboptimal rewards. We find that LLA seems to have slightly improved performance using this smaller architecture, especially over many of the real datasets.

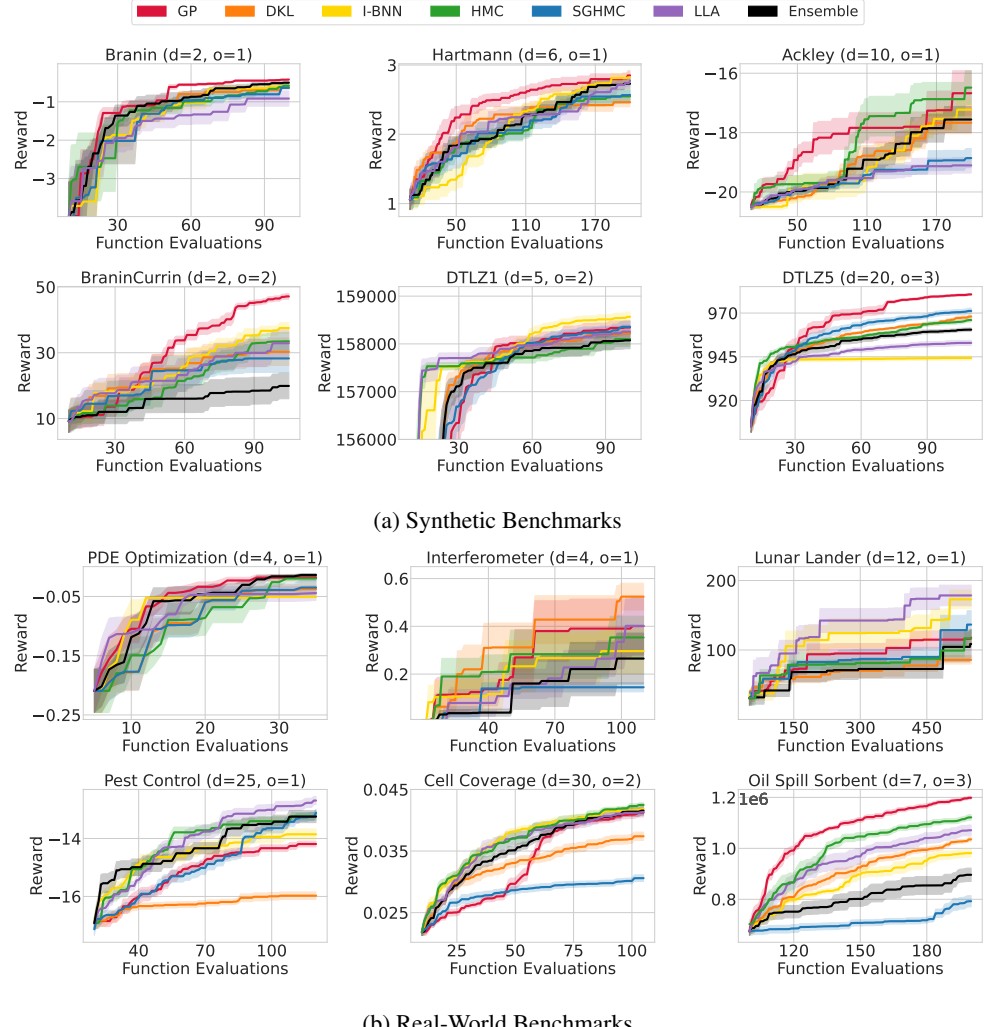

(a) Synthetic Benchmarks

(b) Real-World Benchmarks

Figure A.2: **BNN results with a different architecture**. We show the performance of BNN surrogate models with the relatively small architecture specified by Kristiadi et al. (2023): an MLP with 2 hidden layers of size 50 with ReLU activation. For each benchmark, we include $d$ for the number of input dimensions, and $o$ for the number of objectives. We plot the mean and one standard error of the mean over 10 trials.

## D.2 ABLATION STUDIES

To better understand the quality of the mean estimates and uncertainty estimates of different surrogate models, we conducted ablation studies by creating hybrid models which combine the mean estimate of one surrogate with the uncertainty estimate of another. When we compare this hybrid model with each individual model, we are able to get insight into the relative performance of the mean and uncertainty estimates. We provide the results for GP vs HMC, the gold standard inference for BNNs, and GP vs I-BNN, the model with the most success in high-dimensions.

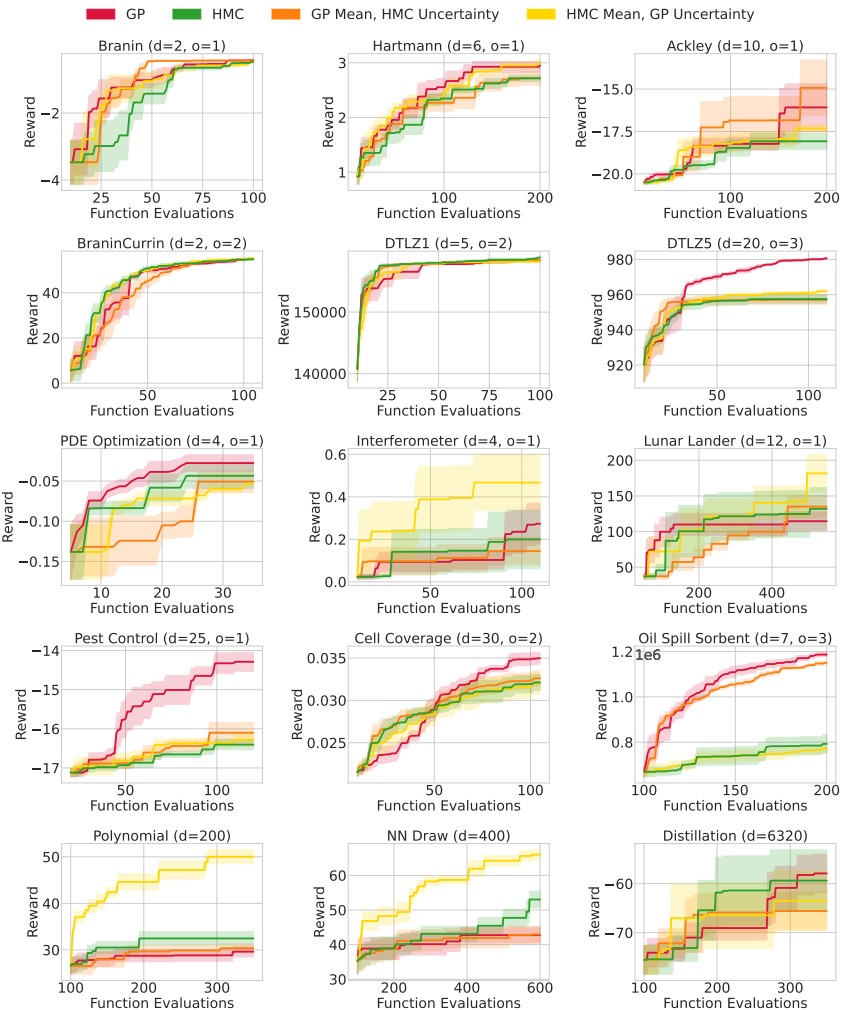

Figure A.3: When we compare GP vs HMC in lower dimensions, we see that mean and uncertainty estimates of the GP need to work together to achieve optimal results, and HMC does not significantly improve when we instead use the GP mean or uncertainty estimates. In contrast, in higher dimensions, the best performing model is the HMC-Mean GP-Uncertainty hybrid, suggesting HMC has better mean estimates while GPs have better uncertainty estimates in this setting. We plot the mean and one standard error of the mean over 10 trials, $d$ refers to the number of input dimensions, and $o$ refers to the number of output dimensions.

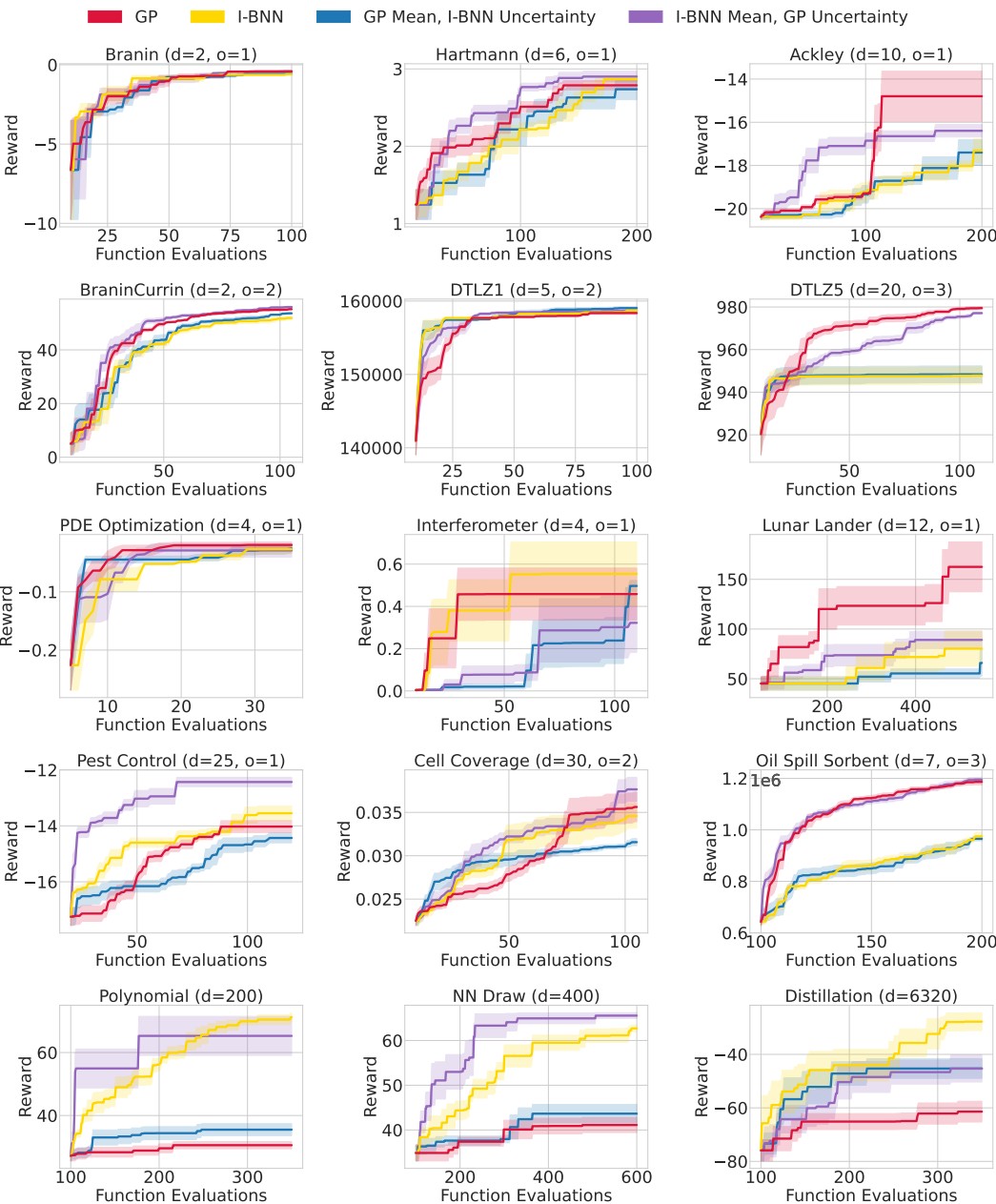

Figure A.4: The I-BNN-Mean GP-Uncertainty hybrid model outperforms GPs across a diverse set of problems, suggesting I-BNNs often provide better mean estimates. In the highest-dimensional problem of Knowledge Distillation (6,320 dimensions), I-BNNs outperform GPs in both the uncertainty and the mean estimates, suggesting that their non-Euclidean and non-stationary similarity metric is advantageous. We plot the mean and one standard error of the mean over 10 trials, $d$ refers to the number of input dimensions, and $o$ refers to the number of output dimensions.

## D.3 DEEP ENSEMBLES

While deep ensembles often provide good accuracy and well-calibrated uncertainty estimates in other settings (Lakshminarayanan et al., 2017), we show they can perform relatively poorly for Bayesian optimization. For instance, when compared to other BNNs on benchmark problems such as BraninCurrin and DTLZ1, the maximum reward found by deep ensembles plateaus at a lower value than other surrogate models. These findings are also supported by results shown in Dai et al. (2022), where deep ensembles do not perform well in Bayesian optimization because they are unable to to explore the space effectively.

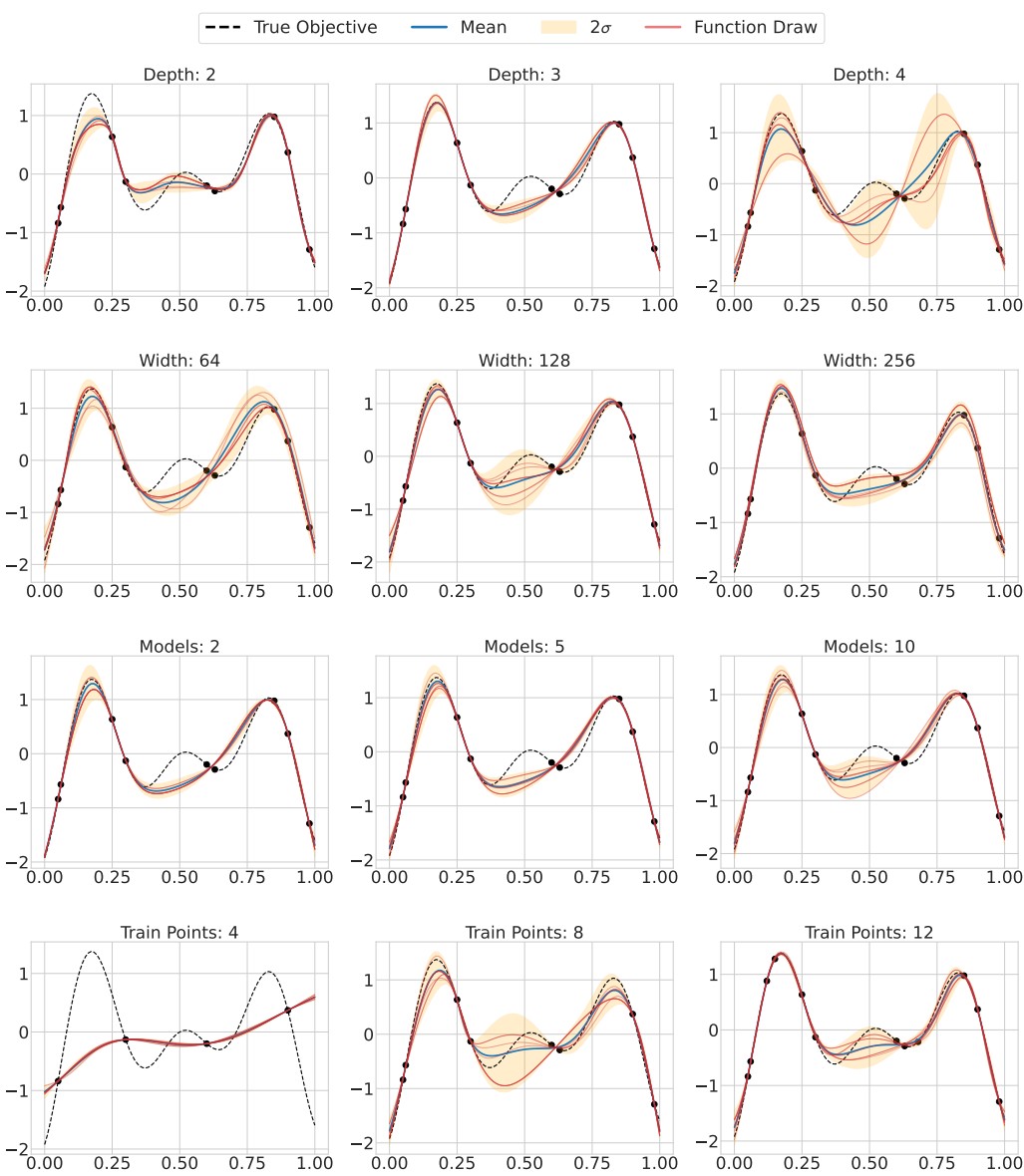

Figure A.5: We visualize the uncertainty of deep ensembles in various scenarios. For each set of experiments, we fix all other design choices with the following base parameters: depth = 3, width = 128, models = 5, train points = 9.

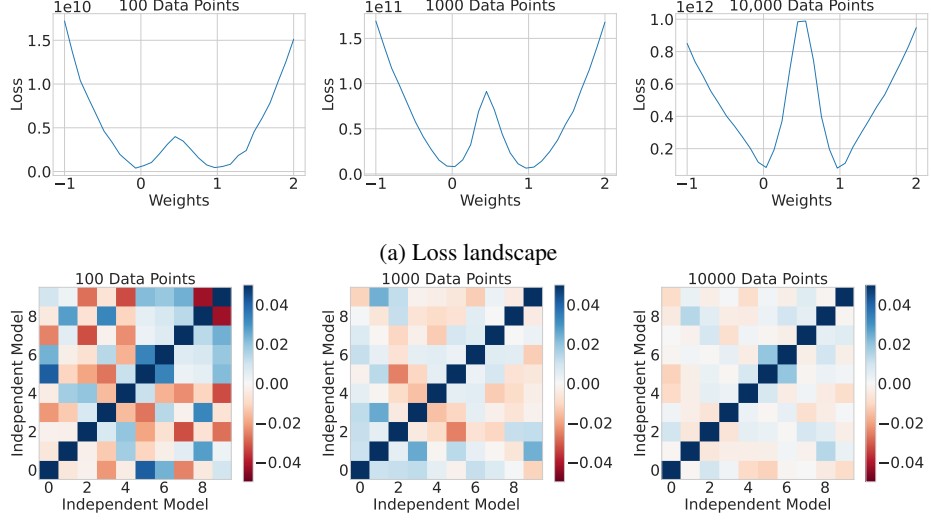

(a) Loss landscape

(b) Cosine similarity between models

Figure A.6: **With minimal training data, the loss landscape is relatively smooth, and separately-trained models are less diverse.** In the *top* figure, we train two models, corresponding to $x = 0$ and $x = 1$. We then linearly interpolate the weights between the two models to measure how the loss changes. As we increase the number of datapoints, the loss landscape becomes less smooth and models are able to find diverse basins of attraction. For the *bottom* figure, we train 10 models and plot the cosine similarity between weights. With less training data, the weights of the models are more related and the models are less diverse. We plot the results for DTLZ1 with 5 input dimensions and 2 output dimensions.

To further investigate the behavior of deep ensembles, we conduct a sensitivity study, varying the architecture, the amount of training data, and the number of models in the ensemble.

In Figure A.5, we see that smaller training sizes can paradoxically lead to less uncertainty with deep ensembles. A critical component in the success of deep ensembles is the diversity of its models. After training, each model falls within a distinct *basin of attraction*, where solutions across different basins correspond to diverse functions. Intuitively, however, in the low data regime there are fewer settings of parameters that give rise to easily discoverable basins of attraction, making it harder to find diverse solutions simply by re-initializing optimization runs. We consider, for example, the straight path between the weights of Model 1 and Model 2, and we follow how the loss changes as we linearly interpolate between the weights. Specifically, given neural network weights $\mathbf{w}_1$ from Model 1 and $\mathbf{w}_2$ from Model 2, and $\mathcal{L}(\mathbf{w})$ representing the loss of the neural network with weights $\mathbf{w}$ on the training data, we plot the loss $\mathcal{L}(\mathbf{w}_2 + (\mathbf{w}_1 - \mathbf{w}_2) * x)$ for varying values of $x$.

We share results in Figure A.6 for DTLZ1, a problem where deep ensembles performed poorly. We can see that in low-data regimes, although the loss between the two models does increase, the loss is significantly higher in other regions of the loss landscape. This behavior suggests that the basins are not particularly distinct, as the loss stays relatively flat between them, and thus less likely to provide diverse solutions. However, as we increase the amount of training data, the models are able to find more pronounced basins.

We also verify that the flatter regions correspond to a decrease in model diversity by measuring the cosine similarity between the model weights, and we see that in the low-data regime, models have more similar weights and therefore are less diverse.

In general, Bayesian optimization problems contain significantly fewer datapoints than where deep ensembles are normally applied. Standard Bayesian optimization benchmarks rarely exceed about 600 data points (objective queries), while in contrast deep ensembles are often trained on problems like CIFAR-10 which have 50,000 training points.

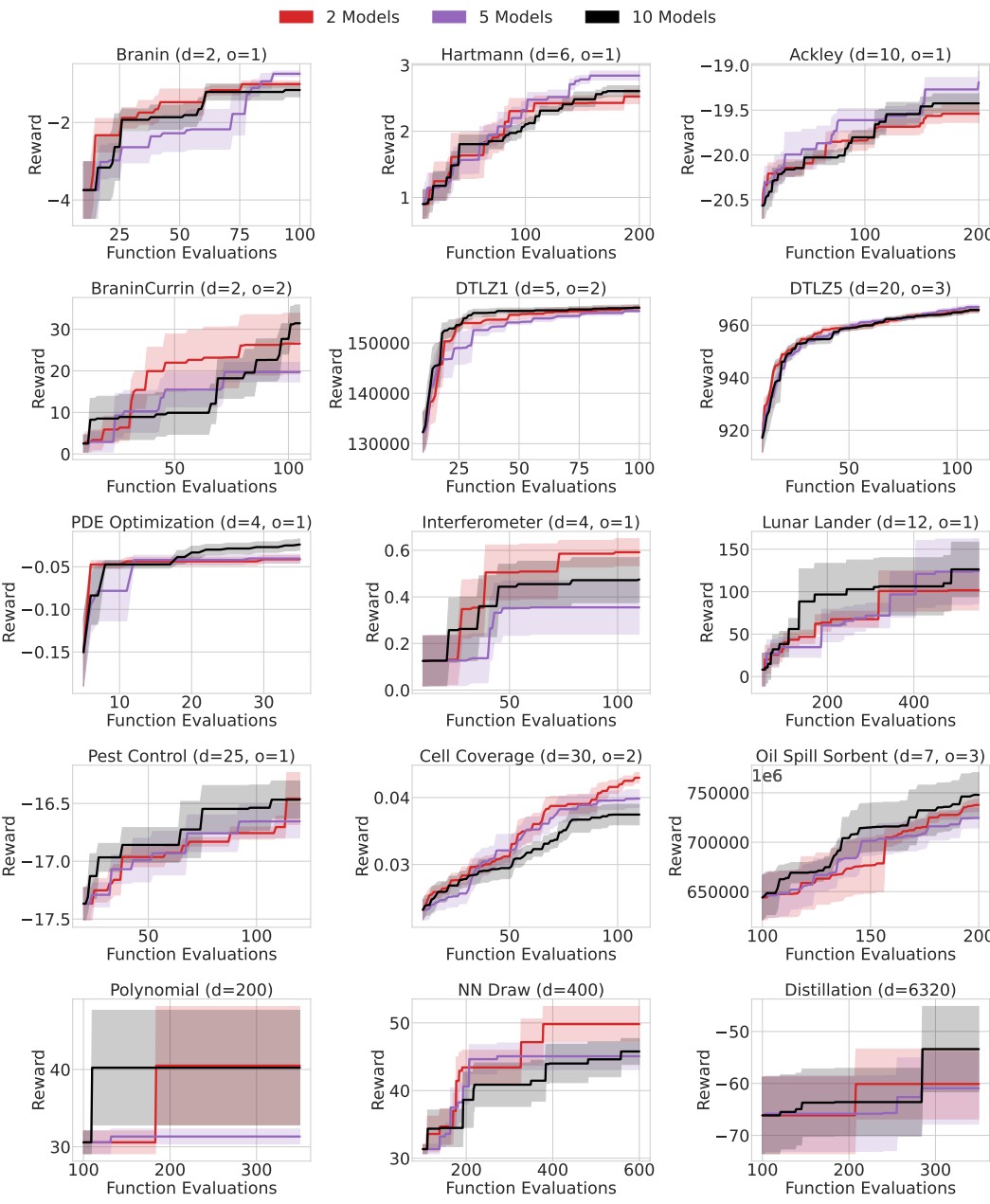

Figure A.7: We compare the behavior of ensembles with different numbers of models, and we find that the different ensembles perform similarly across many experiments, showing the robustness of our results to this hyperparameter. We plot the mean and one standard error of the mean over 10 trials, $d$ refers to the number of input dimensions, and $o$ refers to the number of output dimensions.

## D.4 INFINITE-WIDTH BNNS

I-BNNs outperformed GPs in Bayesian optimization on a variety of high-dimensional problems, and we show results in Figure A.8. The first row of results corresponds to finding the maximum value of a random polynomial function, and the second and third rows show the results of maximizing a function draw from a neural network. For the neural network function draw benchmark, we experimented with many different architectures for the neural network to ensure that we had a diverse set of objective functions to maximize, as denoted in the title of each plot. In all cases, I-BNNs were able to find significantly larger values than GPs.

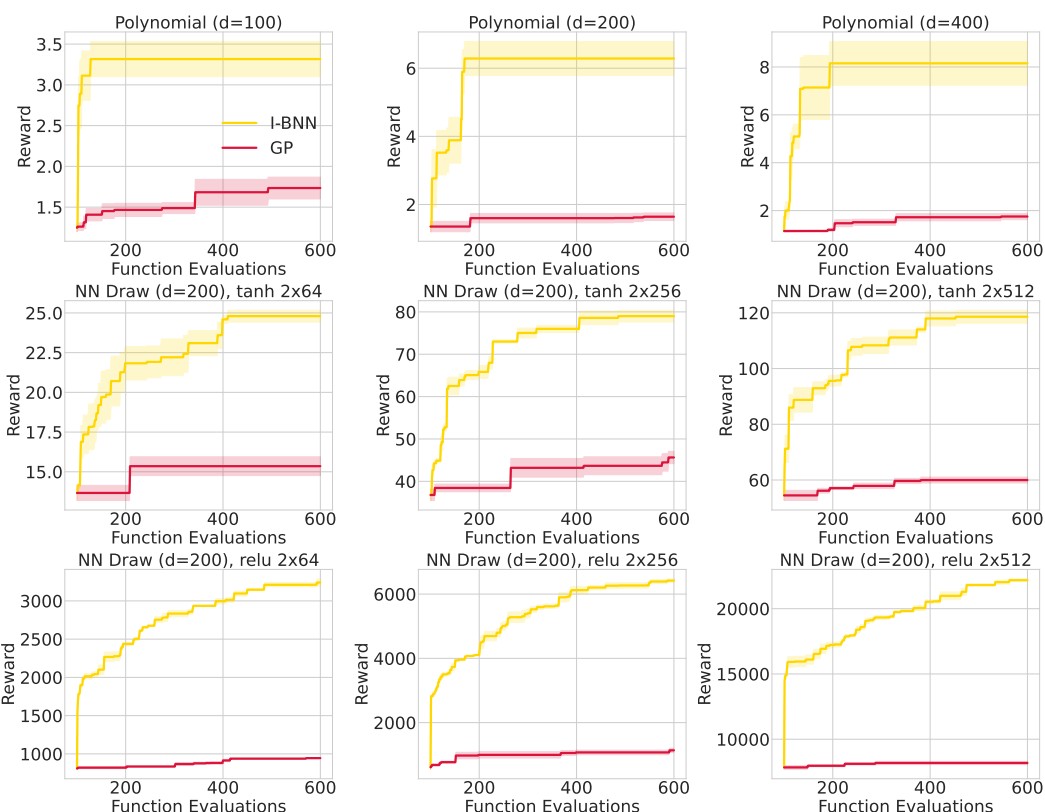

Figure A.8: **I-BNNs outperform GPs in high dimensions.** We plot the mean and one standard error of the mean over 3 trials, and $d$ corresponds to the number of input dimensions. For the neural network draw benchmark, the architecture of the neural network which the function is drawn from is also included, where $2 \times 64$ means the network has 2 hidden layers of 64 nodes each.

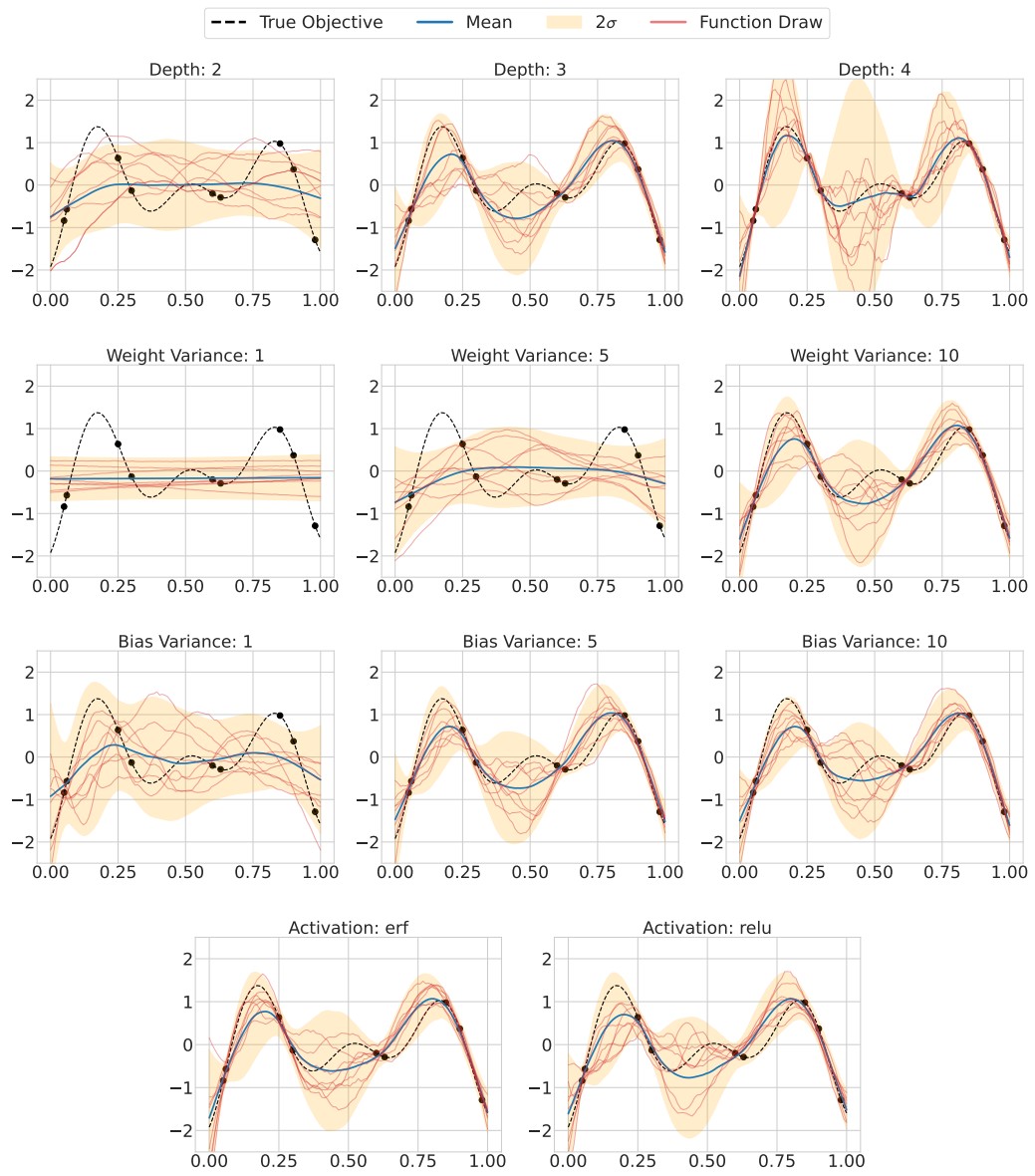

Figure A.9: We visualize how posterior predictive distribution of I-BNNs changes with different design choices. For each set of experiments, we fix all other design choices with the following base parameters: depth = 3, weight variance = 10, bias variance = 5, activation = relu.

In Figure A.9, we conduct a sensitivity analysis on the design of I-BNNs, showing how the posterior changes as we vary certain hyperparameters. Unlike standard GPs with RBF or Matérn kernels which are based in Euclidean similarity, I-BNNs instead have a non-stationary and non-Euclidean similarity metric which is more suitable for high-dimensional problems. Additionally, I-BNNs consist of a relatively strong prior, which is particularly useful in the data-scarce settings common to high-dimensional-problems.

| Problem | Dim | GP MLL | I-BNN MLL |
|---|---|---|---|
| Branin | 2 | -3775.3 | -1354.1 |
| Hartmann | 6 | -1320.4 | -1630.1 |
| Ackley | 10 | -1463.6 | -1824.3 |
| Ackley | 20 | -1657.3 | -2070.2 |
| Ackley | 50 | -2788.9 | -2316.4 |
| Ackley | 100 | -4131.6 | -2456.9 |
| NN Draw | 10 | -3226.9 | -1829.6 |
| NN Draw | 50 | -15544.3 | -2344.7 |
| NN Draw | 100 | -39197.5 | -2453.8 |
| NN Draw | 200 | -35775.2 | -2584.0 |
| NN Draw | 400 | -48710.9 | -2737.2 |
| NN Draw | 800 | -63123.4 | -2868.9 |
| Distillation | 6230 | -2126658.7 | -3059.8 |

Table A.1: **I-BNNs have larger marginal likelihoods than GPs on high-dimensional problems.** Here, we show the marginal log likelihood (MLL) of GPs and I-BNNs on various benchmark problems, and we estimate the MLL by sampling 1000 points randomly from the input domain.

To explore the suitability of the neural network kernel, we compare the marginal likelihood of GPs and I-BNNs on problems with various dimensions, and we report results in Table A.1. We see that I-BNNs and GPs have similar marginal likelihoods on problems with fewer dimensions, such as Branin and Hartmann, but as we increase the number of dimensions, the marginal likelihood of GPs becomes lower than that of I-BNNs. Interestingly, we see that the marginal likelihood of GPs does not decrease as quickly for Ackley as it does for the neural network draw test problem. This may be due to the neural network draw objective function having higher non-stationarity, so the standard GP kernel is less suitable for this problem compared to Ackley. For the knowledge distillation experiment, we see that the marginal likelihood of GPs plummets, and it was also not able to find high rewards for the problem as shown in Figure 5. Overall, I-BNNs have a higher marginal likelihood than GPs on high-dimensional problems, indicating that the I-BNN prior may be more reasonable in these settings.

### D.5   HAMILTONIAN MONTE CARLO

We also explore the effect of the activation function on HMC. In Figure A.10, we see that the function draws from BNNs with tanh activations appear quite different from the function draws from BNNs with ReLU activations. The tanh draws are smoother and have more variation, while the ReLU draws seem to be more jagged. We also include results in Figure A.11 which compare the performance of ReLU and tanh activations in Bayesian optimization problems. We see that there does not appear to be an obvious trend, the optimal choice of activation function is problem-specific.

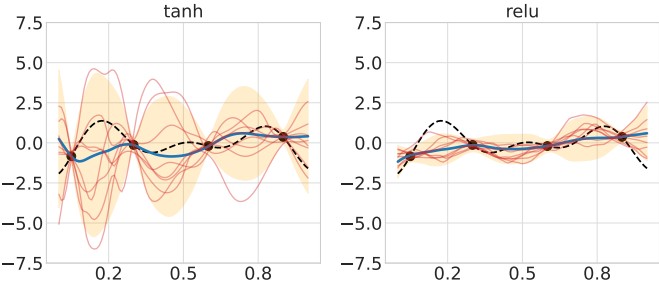

Figure A.10: The function draws of BNNs with tanh activations appear to be similar to the function draws from a GP. In contrast, the function draws from BNNs with ReLU are often jagged, and the network seems to deviate less from the mean.

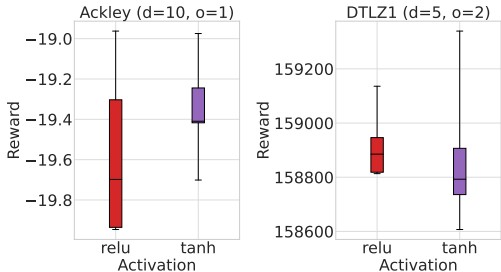

Figure A.11: In practice, ReLU and tanh activation functions can have comparable performance on synthetic functions.

### D.6 GAUSSIAN PROCESSES

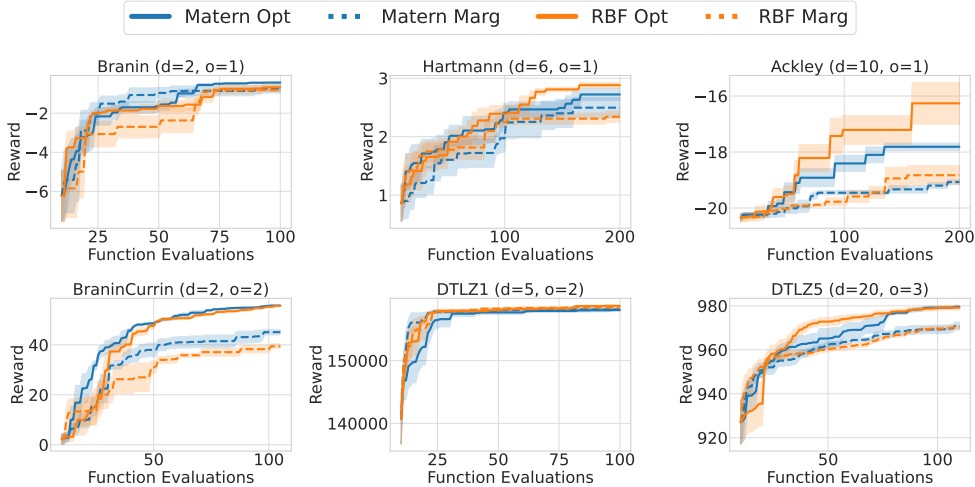

Figure A.12: **The point estimate for hyperparameter selection tends to improve performance on synthetic datasets.** The solid lines refer to hyperparameter optimization using a point estimate, while dashed lines correspond to fully Bayesian marginalization. We see that while the methods perform similarly on Branin, Hartmann, and DTLZ1, the point estimates perform slightly better for other datasets. The optimal choice of GP kernel is also problem-dependent. In many instances, Matérn and RBF perform similarly, although the point estimates for Matern outperform RBF on Ackley. We plot the mean and one standard error of the mean over 10 trials, $d$ refers to the number of input dimensions, and $o$ refers to the number of output dimensions.

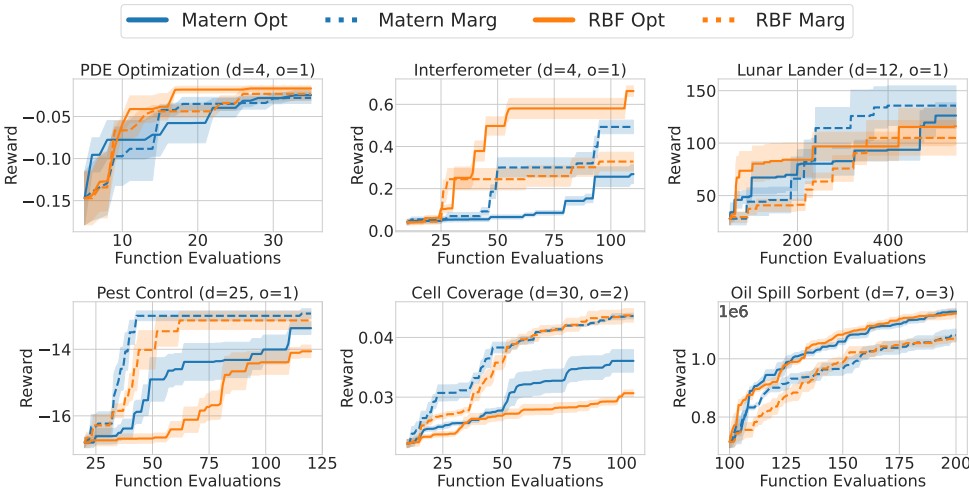

Figure A.13: **The method of hyperparameter selection greatly impacts the performance of GPs on real-world datasets.** The solid lines refer to hyperparameter optimization using a point estimate, while dashed lines correspond to fully Bayesian marginalization. We see real-world datasets often find marginalization of the hyperparameters to be more effective. The optimal choice of GP kernel is also problem-dependent. In many instances, Matérn and RBF perform similarly, although the point estimates for Matérn outperform RBF on discrete problems such as Pest Control and Cell Coverage. We plot the mean and one standard error of the mean over 10 trials, $d$ refers to the number of input dimensions, and $o$ refers to the number of output dimensions.

## D.7 DEEP KERNEL LEARNING

Rather than using the marginal likelihood to optimize parameters for DKL, we can also use the conditional marginal likelihood (Lotfi et al., 2023). We see in Figure A.14 that there is no clear preference for using the marginal likelihood (ML), which we use through all other experiments in the paper, or conditional marginal likelihood (CML) for our problems.

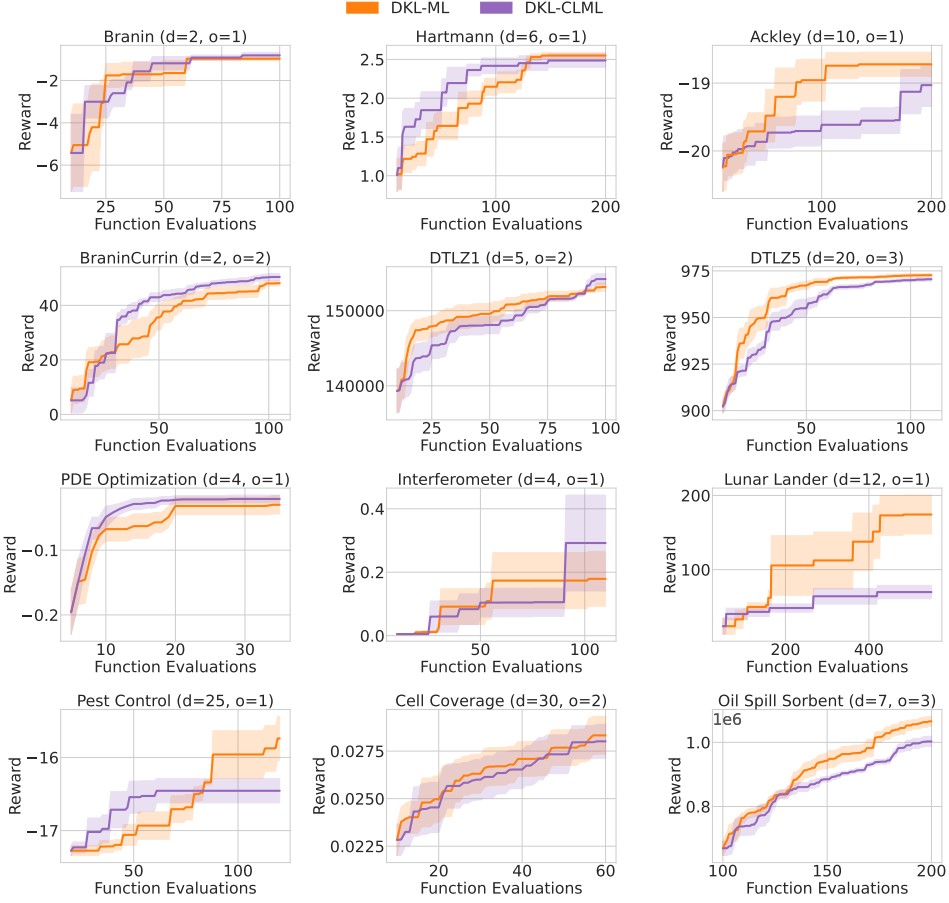

Figure A.14: There is no clear preference for using the marginal likelihood (ML) or conditional marginal likelihood (CML) to optimize the parameters of DKL

## D.8 ACQUISITION BATCH SIZE

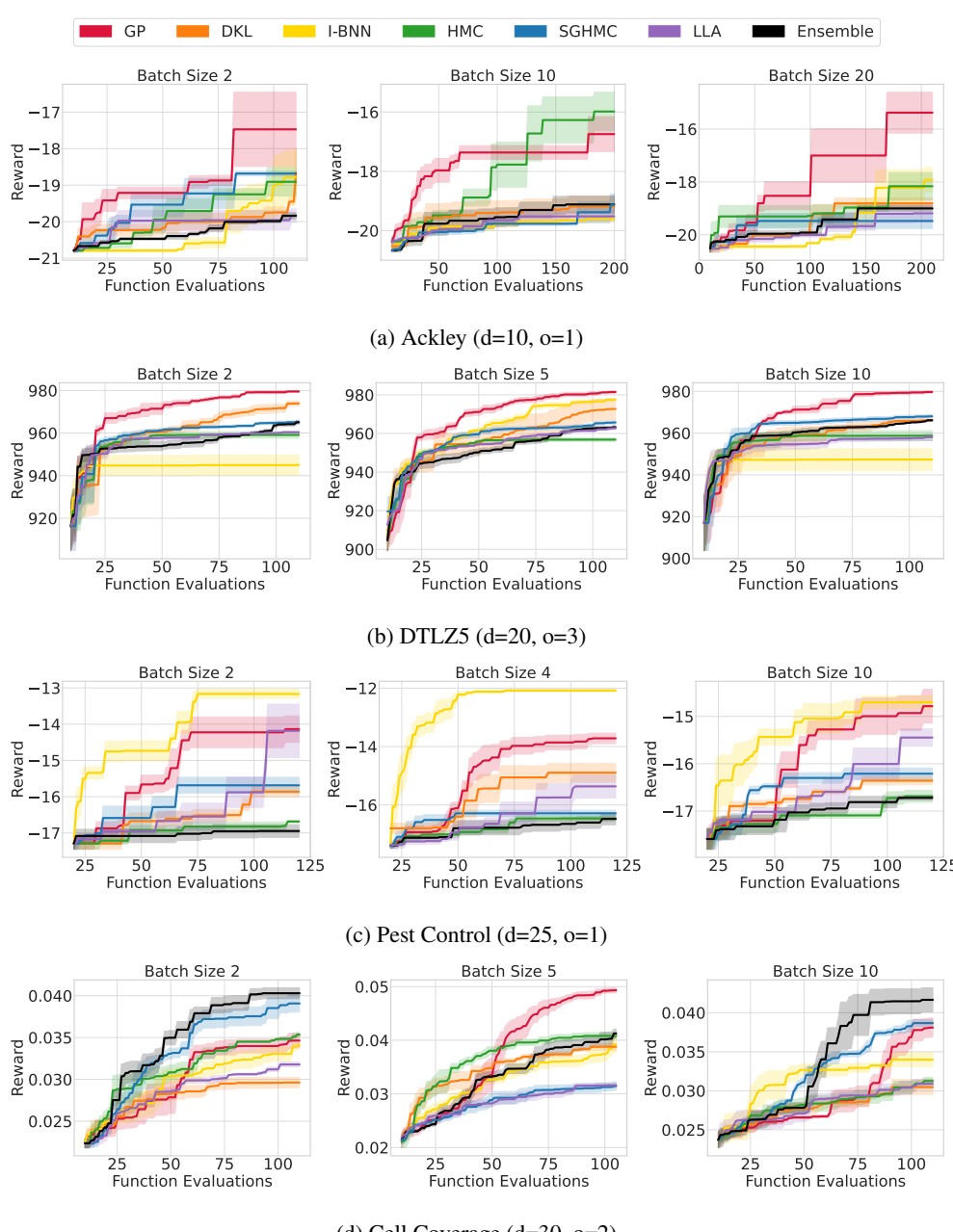

(a) Ackley (d=10, o=1)

(b) DTLZ5 (d=20, o=3)

(c) Pest Control (d=25, o=1)

(d) Cell Coverage (d=30, o=2)

Figure A.15: The batch size of the acquisition function impacts the performance of the models, but the general trends remain similar. The relative performance of the models are similar across varying batch sizes, with a few notable exceptions. The I-BNN seems to plateau on DTLZ5, and is unable to find high rewards. GPs also seem to perform significantly better on Cell Coverage with a batch size of 5 compared to other sizes. For each benchmark, we include $d$ for the number of input dimensions, and $o$ for the number of objectives. We plot the mean and one standard error of the mean over 10 trials.

## D.9 LIMITATIONS OF GAUSSIAN PROCESS SURROGATE MODELS

Due to their assumptions of stationarity as noted in Section 4.3, GPs struggle in non-stationary settings. In Figure A.16, we compare the posterior distribution from a GP with the posterior from different BNN surrogate models. We show results after 20 iterations of Bayesian optimization, starting with an initial point of $x = 0$. The function we wish to maximize is non-stationary: the function has greater variance between $-2$ and $2$, and there is also a slight downward trend. We see that due to their stronger assumptions, GPs are not able to find the true global maximum of 0.8, instead getting suck in local optima. In contrast, HMC and I-BNNs are able to find the global maximum within 20 iterations.

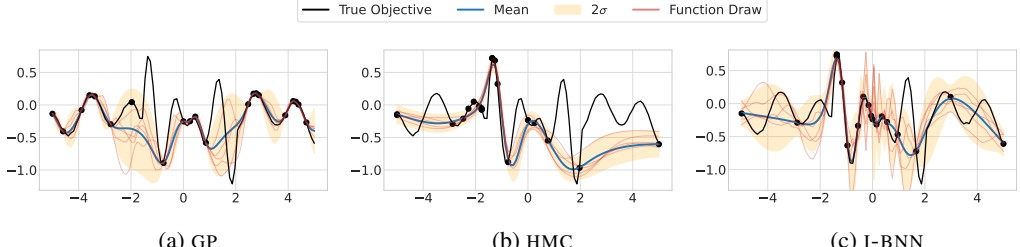

(a) GP            (b) HMC            (c) I-BNN

Figure A.16: **GPs struggle to find the global maximum for non-stationary functions**. After 20 function evaluations, the GP does not accurately model the true function around $x = -1$ because it is unable to account for the sudden increase in scale due to its assumption of stationarity. BNNs do not suffer from the same pathologies and are able to find the true maximum.

An additional limitation of GP surrogate models, which we demonstrate in Figure A.17, is its performance on multi-objective problems in Bayesian optimization. Although GPs have been successfully extended to a wide range of multi-objective problems, in the interest of making the approaches scalable, there are many assumptions placed onto the kernel. In the most naive setting, we can model each objective independently. While this approach is convenient, it completely ignores the relationship between objective values and has no notion of shared structure, so it is unable to take advantage of all of the information in the problem. Multi-objective covariance functions can also be decomposed as Kronecker product kernels. While this approach can have significant computational advantages compared to modeling the full covariance function, it requires each objective to itself be modeled with the same underlying kernel. Thus, this method of modeling multiple objectives will fail to capture the nuances of each particular objective when the functions have differing properties.

In Figure A.17, we show the result of twenty iterations of Bayesian optimization over a synthetic multitask example, where we care about optimizing over the fourth function but provide additional information through the other three datasets. We use the GP with Kronecker product kernels to model the multiple objectives. In our experiment, although the GP is able to learn the proper length scale and variance over the three additional functions with similar length scales, it struggles to accurately fit the fourth objective. Because the GP is unable to account for the differences between the four functions, it does not find the global optimum. Unlike GPs, BNNs are not restricted to strict covariance structures and are able to produce well-calibrated uncertainty estimates in multi-objective settings. The BNNs are able to accurately fit all four functions, including the fourth objective function which has a much smaller length scale compared to the others.

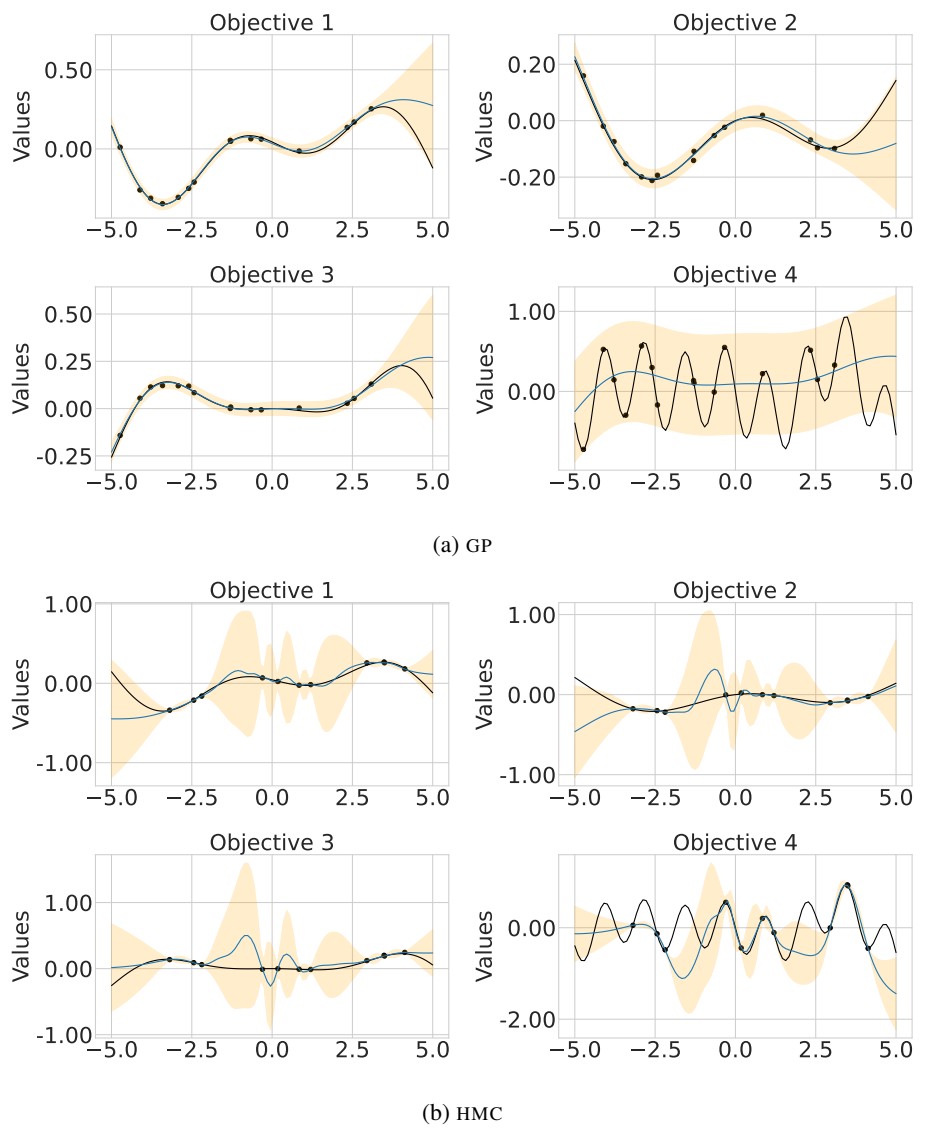

Figure A.17: **GPs have a hard time finding the global maximum in multi-objective settings**. Multi-task GPs learn one length scale across all objectives, which may not be suitable for many datasets. In this example, it does not find the global minimum in the 4th objective because it treats the shorter length scale as noise. In contrast, BNNs are able to appropriately model the uncertainty for the 4th objective and find its true minimum.

## D.10 LARGE NUMBER OF FUNCTION QUERIES

To further accentuate the distinctions between BNNs and GPs, we experiment with a larger number of function queries. Specifically, we look to maximize a function with 200 input dimensions drawn from a fully connected neural network with 5 layers and 256 nodes per layer. We expect this function to have a high degree of non-stationarity, making it difficult for standard GPs. We start with 2000 initial function queries and end the experiment at 3000 function queries. We share results in Figure A.18.

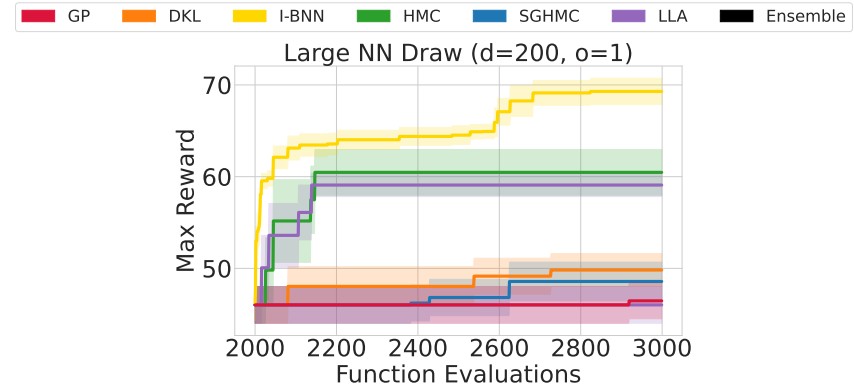

Figure A.18: BNNs outperform GPs when there are a large number of function queries.

This experiment reveals several valuable findings: (1) I-BNNs remain competitive relative to the alternatives; (2) BNN surrogates start to perform more effectively, able to leverage the additional data for representation learning; (3) deep ensembles, in particular, are greatly improved with more data, in line with our explanation that the relative poor performance was due to an inability to find diverse models corresponding to posterior modes when there is limited data. At the same time, we note most Bayesian optimization problems do not have many objective queries, since Bayesian optimization is often found to be most valuable when the objective is expensive to query. In this light, perhaps (1) is the most valuable of the findings, since it shows consistency of the relatively strong I-BNN surrogate. In the future, we might expect I-BNNs to become a mainstream surrogate model for Bayesian optimization.

## D.11 RUNTIME

While inference time is relevant for the comparison of surrogate models, in many real-world Bayesian optimization scenarios, the most expensive computation often lies in the querying of the objective function, which may include actions such as synthesizing a new material, training a large neural network to convergence, etc. For these scenarios, the quality of the surrogate model uncertainties may be much more important than the cost of inference. For completeness, in rare instances where inference-time is a relevant consideration, we provide wall-clock times of all surrogate models across our experiments in Table A.2, and we compare the performance of the surrogate models within a fixed time budget in Figure A.19

| Problem | GP | DKL | I-BNN | HMC | SGHMC | LLA | Ensemble |
|---|---|---|---|---|---|---|---|
| Branin | 2.62 | 124.55 | 3.97 | 214.40 | 142.21 | 19.29 | 190.24 |
| Hartmann | 8.20 | 123.00 | 7.79 | 663.22 | 381.57 | 49.25 | 413.50 |
| Ackley | 4.28 | 263.78 | 4.69 | 141.66 | 201.48 | 28.29 | 225.41 |
| BraninCurrin | 14.17 | 198.80 | 19.76 | 237.71 | 365.86 | 61.09 | 432.94 |
| DTLZ1 | 8.25 | 117.02 | 10.82 | 183.74 | 127.54 | 21.97 | 194.68 |
| DTLZ5 | 21.50 | 308.12 | 22.18 | 194.81 | 147.07 | 38.99 | 231.96 |
| PDE | 240.48 | 512.98 | 239.23 | 456.48 | 628.14 | 312.40 | 981.50 |
| Interferometer | 16.09 | 939.49 | 17.97 | 690.07 | 540.59 | 83.61 | 944.87 |
| Lunar Lander | 458.96 | 3140.30 | 621.20 | 3118.65 | 684.61 | 802.13 | 829.44 |
| Pest Control | 3.84 | 268.06 | 5.43 | 66.18 | 219.08 | 27.87 | 253.37 |
| Cell Coverage | 9.50 | 593.09 | 13.26 | 101.23 | 181.11 | 35.00 | 243.17 |
| Oil Spill | 37.51 | 300.20 | 44.81 | 291.53 | 350.30 | 124.71 | 438.89 |
| Polynomial | 16.28 | 673.40 | 20.59 | 211.94 | 486.75 | 106.77 | 540.41 |
| NN Draw | 232.48 | 1458.11 | 391.95 | 896.34 | 451.34 | 799.88 | 1124.63 |
| Distillation | 4719.40 | 3982.80 | 4937.27 | 4282.89 | 299.40 | 3223.16 | 3007.93 |

Table A.2: Wall-clock time in seconds of one trial of each experiment, with experiment details specified in Appendix C. We record the mean time over 10 trials.

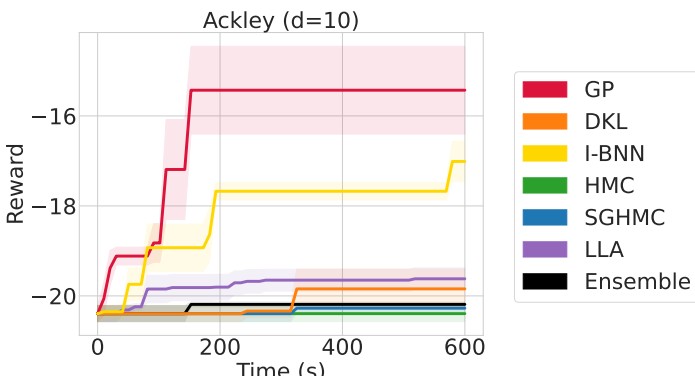

Figure A.19: For objective functions which are very inexpensive to query, like Ackley, we find GPs and I-BNNs to outperform other surrogate models given a fixed amount of time. However, for the many Bayesian optimization problems instead which expensive to query, and the total time would be dominated by the function query instead of inference time. We record the mean and standard error over 10 trials.

