# OpenReview forum: "A Study of Bayesian Neural Network Surrogates for Bayesian Optimization"
_ICLR.cc/2024/Conference — ICLR 2024 poster_

### Official Review · Reviewer_v5S3 · 2023-10-22

**Soundness:** 3 good
**Presentation:** 2 fair
**Contribution:** 1 poor
**Rating:** 6
**Confidence:** 4

**Summary:**

This paper creates a study about the performance of Bayesian surrogates inside the Bayesian optimization framework. They performed experiments on real and synthetic datasets in order to understand the performance of surrogates. They report some findings such as the ranking of methods is highly problem dependent, HMC is the most successful approximate inference procedure for fully stochastic BNNs, and that infinite-width BNNs are promising.

**Strengths:**

- The topic is interesting and somewhat important for the community.
- The presentation of theory and the experiments is well presented and sound.

**Weaknesses:**

- There is no discussion on the interaction of the surrogate and the acquisition function.
- I agree with the authors that the time might not be relevant when the function evaluations are expensive, still it is important to create an experiment assuming fast function evaluations and see whether the ranking holds
- Although the dataset collection is diverse, the study is performed on a very small amount of datasets. There is no guarantee that these findings extrapolate easily to new datasets.
- Although the insights are interesting (some of them are not surprising), the effective impact is questionable. How could these insights lead to SotA in drug discovery algorithms, active learning, material science or hyperparameter optimization?
- No discussion on the regularization effect on the surrogates. What happens if I regularize the DKL or the Bayesian neural networks via some useful prior?

**Questions:**

- How would more recent methods rank in the comparison, such as PFN4BO [1]?
- How is the performance of DKL affected by regularization approaches  such as [2], [3]?


[1] Müller, Samuel, et al. *PFNs Are Flexible Models for Real-World Bayesian Optimization.*

[2] Lotfi, S., Izmailov, P., Benton, G., Goldblum, M., & Wilson, A. G. *Bayesian model selection, the marginal likelihood, and generalization.*

[3] Patacchiola, M., Turner, J., Crowley, E. J., O'Boyle, M., & Storkey, A. J. *Bayesian meta-learning for the few-shot setting via deep kernels.*

---

> ### Author Response · Authors · 2023-11-20
> **Author Response to Reviewer v5S3 [1/2]**
>
> We appreciate your review, and we respond to your questions as follows.
>
> > Importance of insights
>
> The choice of surrogate model in Bayesian optimization is a fundamental design decision that has been largely overlooked in the literature, and many papers default to using standard Gaussian process surrogates (e.g., with Matern or RBF kernels). Our paper is the first to provide a comprehensive study of BNN surrogates, and the wide range of neural network-based surrogates explored in our submission allows us to evaluate the role of representation learning, non-stationarity, and stochasticity in modeling Bayesian optimization objectives. These findings offer pragmatic insights for BO practitioners by providing guidance on selecting appropriate surrogate models.
>
> Our paper also has many novel and important findings, in some cases even prompting us to re-think conventional wisdom in BO. Surprising and unexpected findings include the following: (1) There are limited empirical benefits of stochasticity in the surrogate, given the competitive performance of DKL to BNN models; (2) the performance of deep ensemble surrogates is surprisingly poor, despite their success in other non-BO contexts; (3) I-BNN demonstrate compelling performance on high dimensional benchmarks; (4) Different BO objectives exhibit highly different salient structures (e.g., in contrast to many vision and NLP problems). We additionally note that we are, to the best of our knowledge, the first to consider infinite BNN and HMC surrogates, and I-BNNs have strikingly good results in the high dimensional settings. While not every result would be expected to be surprising, many of the results in our submission are both surprising and informative.
>
> In this paper, we do not try to develop new methods for specific Bayesian optimization applications and note that in order to reach state-of-the-art performance in drug discovery, materials engineering, or some applied problem of that nature, the paper would need to be entirely about that single applied problem and the idiosyncrasies associated with it. Our focus is instead on a general foundational scientific study on the effectiveness of BNN surrogate models on a wide range of problems, covering a wide array of standard benchmarks, with various different properties. We have further outlined the contributions of the paper in the separate general post.
>
> > Number of Datasets
>
> We respectfully disagree that we consider a "very small number of datasets". We benchmarked our results on 15 different datasets, which is more than many other related Bayesian optimization papers:
> - 8 datasets: Promises and Pitfalls of the Linearized Laplace in Bayesian Optimization, Kristiadi et al. 2023
> - 5 datasets: Scalable Global Optimization via Local Bayesian Optimization, Eriksson et al. 2019
> - 8 datasets: Scalable Bayesian Optimization Using Deep Neural Networks, Snoek et al. 2015
> - 3 datasets: Maximizing Acquisition Functions for Bayesian Optimization, Wilson et al. 2018
> - 9 datasets: Bayesian Optimization with High-Dimensional Outputs, Maddox et al. 2021
> - 11 datasets: Bayesian Optimization with Robust Bayesian Neural Networks, Stefan et al. 2016
>
> > What happens if I regularize the DKL?
>
> We explored this question in the submitted manuscript. In Appendix D.7, we include results for DKL as suggested in [2], showing the performance of optimizing the parameters with the marginal likelihood (ML) compared to the conditional marginal likelihood (CML). We see that there is no clear preference for using the ML or the CML, and the behavior of DKL with ML and CML is very similar across many of the objectives.
>
> > Runtime of Bayesian Optimization
>
> We provide wall-clock times of the surrogate models across our experiments in Appendix D.11. However, we would like to emphasize that Bayesian optimization applications often include expensive computation querying of the objective function, which may include actions such as synthesizing a new material, training a large neural network to convergence, etc. For these scenarios, the quality of the surrogate model uncertainties may be much more important than the cost of inference.
> That said, all of our runtimes are available, and we are not advocating for one method over another in general. Practitioners can use the runtimes we provide as a guide in making a decision for what matters most to them in the context of their problem. We also note that I-BNNs, a novel surrogate model in our paper, has both relatively fast runtime, and strong results.
>
> Edit: See new comment below! We have provided a new experiment in Appendix D.11 to address this point.

---

> ### Author Response · Authors · 2023-11-20
> **Author Response to Reviewer v5S3 [2/2]**
>
> > Extensive Experiments
>
> Our paper includes a diverse set of experiments showing how many different aspects of Bayesian optimization interact with our results. For each of our surrogate models, we include experiments in the appendix where we vary their hyperparameters, such as the number of models in a deep ensemble or the kernel for a GP, and measure their impact on performance. Additionally, we also include experiments showing the impact of different BNN architectures, and we also show the effects of the batch size of the acquisition function. While it is always possible to request additional experiments, we believe that we have done an exhaustive and extensive exploration of BNNs for Bayesian optimization.
>
>
> > How would recent methods such as PFNs4BO rank?
>
> Thank you for the reference! Our paper focuses on the performance of Bayesian neural networks trained from scratch, and consequently, we find methods reliant on pre-training to be outside the scope of our work. The inclusion of methods not trained from scratch makes an apples-to-apples comparison especially difficult, since many of the other neural net surrogates could also be pre-trained in some way. However, we are excited about methods that leverage auxiliary data, which we believe may be a notable advantage for using neural network surrogates for Bayesian optimization. We have included a reference to PFNs4BO in the updated manuscript and included it in our discussions.

---

> ### Author Response · Authors · 2023-11-21
> **Author Response to Reviewer v5S3, Runtime Experiment**
>
> In response to your comment on runtime, we have now updated Appendix D.11 with a new experiment comparing the performance of the surrogate models within a time budget, under the assumption of very fast function queries. We find that in this setting, GPs and I-BNNs outperform other BNNs which have slower inference times. However, we would like to re-emphasize that this scenario is often not representative of many common Bayesian optimization problems, which instead have very expensive function queries which would dominate the time budget.
>
> We hope we have addressed your concerns, and please let us know if you have any remaining questions.

---

> ### Author Response · Authors · 2023-11-22
> **Author Response to Reviewer v5S3**
>
> Hello, the rebuttal period will end in about 12 hours! Please let us know if you have any further questions we can address.

---

> > ### Comment · Reviewer_v5S3 · 2023-11-23
> > **Reply to authors**
> >
> > I thank the authors for the response and acknowledge that I read it.

---

### Official Review · Reviewer_XqQk · 2023-10-29

**Soundness:** 3 good
**Presentation:** 4 excellent
**Contribution:** 3 good
**Rating:** 8
**Confidence:** 3

**Summary:**

The paper is an empirical investigation into the use of Bayesian neural networks (BNNs) as surrogate models for BO instead of the traditional GPs. The BNN inference procedures investigated are HMC, SGHMC, deep ensembles, infinite-width BNNs, linearlized Laplace approximations, and deep kernel learning. The authors compare the performance of GPs and the various BNN inference procedures via the maximum reward attained over several standard synthetic benchmarks, real-world benchmarks, and high-dimensional settings. The paper also investigates several secondary aspects of BNNs for BO, including the role of the NN hyperparameters, the performance of hybrid models, the effect of the number of function evaluations available, and the computational runtime.

**Strengths:**

1. The results are of interest to BO researchers and BO practitioners looking for potential methods of improving the effectiveness of BO in real-world applications.
2. The empirical investigation is extensive and carefully planned, covering several synthetic and real-world benchmarks, and provides support for many interesting hypotheses as well, such as the relative performance on high dimensional problems and the role of hyperparameters including network architecture.
3. This paper is a gold standard for clarity and writing.

**Weaknesses:**

1. For an empirical paper whose conclusions rest solely on the experimental results, 5 trials for each experimental setup is too little, as evidenced by multiple plots having heavily overlapping confidence intervals, Figure 6 in particular.

2. A few clarifying questions, please see the Questions section.

**Questions:**

1. This question concerns the experimental details outlined in Appendix C.1 and what I've gleaned from the code. When a GP model is used, it undergoes hyperparameter optimization via maximizing the marginal likelihood w.r.t. to the hyperparameters at every BO iteration. When a HMC model is used, it is also described to undergo a hyperparameter optimization procedure which is an iterated grid search that chooses the set of hyperparameters that (to my understanding) maximizes the maximum reward attained after all BO iterations in a trial. This optimization procedure is different from the GP one in that it requires an entire BO trial to compute the score of a single set of hyperparameters, and hence requires several BO trials as opposed to the GP one that is optimized per iteration and does not require several BO trials. Is this an accurate understanding? If so, could you comment on the validity of comparing the results of the GP model and the HMC model (along with the other BNN models since their hyperparameters are arrived at via the HMC search as well)? The concern is that the GP model did not have the same opportunity to do a 'meta-optimization' over several BO trials which might have been used to optimize other (hyper-)hyperparameters such as the choice of lengthscale and outputscale priors. Or was it the explicit intention to compare a standard GP setup against an (a priori unknown) 'optimal' BNN ?

2. From Figure 6, GP and I-BNN are the top 2 performing models on all problems, and I-BNN dominates by far on high dimensional problems. However, I-BNNs are equivalent to (and implemented as) GPs with a specific neural-network based kernel. One may reach the conclusion that GPs still reign supreme in BO: use a standard Matern kernel for low dimensional problems, and the I-BNN kernel for high dimensional problems, and ignore BNNs that are not also GPs. Would you say this is a fair alternate conclusion?

---

> ### Author Response · Authors · 2023-11-20
> **Author Response to Reviewer XqQk [1/2]**
>
> We really appreciate your review and thoughtful comments. Below we respond to your questions, and include several substantial experiments inspired by your comments. We also have a general response, summarizing our contributions.
>
> > Question 1: GP vs HMC experimental details
>
> It is a fair point that our process for optimizing HMC differed from our process for setting GP hyperparameters. Indeed we were interested in understanding the potential of BNNs when the hyperparameters are well-specified for a particular problem.
>
> Inspired by your comments, we have now conducted additional experiments for HMC where we optimize the hyperparameters on a per-iteration basis, similar to the procedure for GPs. Due to time constraints, we have focused on a subset of the objective functions, but we will include a full set of results across all problems in the final paper. Specifically, we fix the width and depth of the network across all experiments, and we conduct a small grid-search after each iteration to optimize the prior and noise hyperparameters based on the validation likelihood, and we include more details about the experiment setup and results in Appendix D.12.
>
> Overall, we find the performance of per-iteration optimized HMC to be competitive with per-trial optimized HMC for problems like Hartmann and Cell Coverage. Furthermore, although per-trial optimization for HMC does improve its performance on Ackley, we are still able to observe similar trends as before.
>
> Furthermore, to make the HMC results reported in the submission more comparable to the GP results, we have also conducted additional experiments where we optimize GP hyperparameters on a per-trial basis. Specifically, we use grid search to select the parameters of the prior distribution of the length scale and output scale for a GP, and for each objective, we choose the parameters which lead to the highest maximum reward over one trial.
>
> We see that these optimized GPs are often comparable to the default GP and do not see significant improvements. This may be due to the GPs being optimized over one specific trial of Bayesian optimization, and the optimal GP may vary depending on the set of initial points. This shows that the observations of the submission hold for both the default GP as well as the meta-optimized version.
>
> We thank you again for the question, and we believe the added experiments, inspired by your comments, will provide significant additional value to the paper.
>
> > Question 2: Do GPs still reign supreme?
>
> Our results show that GPs with a standard Matern kernel works well for lower-dimensional problems and GPs with the I-BNN kernel work well in higher dimensions.
>
> Indeed, GPs encompass a broad class of models. GPs with I-BNN kernels exhibit very different behavior from standard GPs with a Matern or RBF kernel. The infinite-width BNN has a non-Euclidean and non-stationary similarity metric, which enables its relative success in high dimensions. DKL based GPs are also significantly different, with their own properties, such as representation learning.
>
> For this reason, we have been careful to distinguish "standard GPs"---GPs with the kernels almost always used in Bayesian optimization---from other model classes, as GPs allowing for any kernel are so general it becomes difficult to make informative conclusions or comparisons. Indeed, the line can get blurry---I-BNNs are a particular type of GP derived from a BNN.
>
> We also note that we have several nuanced take-aways, rather than a single conclusion, and are careful not to proclaim standard GPs or BNNs as overall winners or losers. There is no desire to show that BNNs are generally superior, and we are in fact careful to highlight the broad appeal of standard GP surrogates. Our goal is to understand how various properties of the surrogate (stochasticity, representation learning, etc.) interact with performance in different settings, which transcend the GP vs. BNN distinction, though many of our considered surrogates are NN-inspired (HMC, SGHMC, Deep Ensembles, I-BNN, DKL). If we define BNNs narrowly, and GPs broadly, it may be reasonable to say that GPs have an edge, but we are most concerned with the low-level property distinctions of the surrogates.

---

> ### Author Response · Authors · 2023-11-20
> **Author Response to Reviewer XqQk [2/2]**
>
> > 5 trials is too little
>
> With five trials we do see sufficient evidence to draw many informative conclusions. However, we agree more trials would be helpful and will run all of our experiments for a minimum of ten trials in the final paper to further differentiate the performance between surrogate models.
>
> Although Figure 6 has heavily overlapping confidence intervals, running more trials may not decrease the width of the intervals. This figure plots the distribution of the relative performance of each surrogate model across all trials and all objective functions. For example, we see that for at least 25% of the trials, the relative performance of GPs is less than 0.5, meaning the maximum reward found by GPs for that trial was closer to the reward found by the worst surrogate than the reward found by the best surrogate. We would expect this ratio to stay the same even as we increase the number of trials.
>
> Thank you again for the thoughtful feedback, which we believe has improved the paper. We have added substantial new experiments inspired by your questions, and would appreciate if you could consider raising your score in light of our response.

---

> > ### Comment · Reviewer_XqQk · 2023-11-21
> >
> > Thanks for your response and for the effort made to address the questions. I have no further concerns and have increased my score.

---

### Official Review · Reviewer_6qj4 · 2023-11-01

**Soundness:** 3 good
**Presentation:** 3 good
**Contribution:** 3 good
**Rating:** 8
**Confidence:** 4

**Summary:**

This paper provides a comprehensive empirical study of using Bayesian neural networks as the surrogate in Bayesian optimization. The paper considers a number of different BNNs, and performed experimental comparisons in a variety of experiments. Some interesting insights are shown from the experiments, including when standard GP surrogate is better and when BNN is better, HMC is often the best method for inference for BNN, deep kernel learning is usually competitive and deep ensemble is usually not, etc.

**Strengths:**

- The methods under comparison are carefully selected to span a wide range of possible BNNs, and experiments are nicely designed to unveil specific insights about the relative strengths/weaknesses of different families of methods.
- I think some of the conclusions/insights from the empirical comparisons can indeed be useful for future applications of Bayesian optimization, such as the competitiveness of deep kernel learning, the promising results of infinite-width BNNs in high-dimensional problems (which is a new observation to the best of my knowledge), etc.
- The synthetic experiments in Figures 1 and 2 are nicely designed to illustrate the influence of different factors, and also find which are the parameter combinations likely to work better. The experiments "Quality of Mean and Uncertainty Estimates" on the potential of mixing different mean and uncertainty estimates are also particularly interesting.
- The paper is well written, the contributions are nicely organized and discussed.

**Weaknesses:**

**(1)** I think it would make the study more complete if another relevant line of works is discussed: using (non-Bayesian) neural networks as the surrogate in BO and using neural tangent kernel for exploration. The recent line of work on neural bandits has made it possible to use (non-Bayesian) neural networks as the surrogate in BO while still preserving the regret guarantee of BO by using the theory of the NTK, The relevance of neural bandits in BO has been shown by [1] below, and you can also refer to [1] to find the related works on neural bandits. In fact, I think the findings in [1] can be used to corroborate some of the findings in this work. For example, [1] also found that deep ensemble doesn't work well and explains it by arguing that deep ensemble cannot do principled exploration, I think this is in fact consistent with what's observed in this work, because the performance of deep ensemble plateaus at low objective values because it's subpar exploration ability makes it unable to find the region containing the global optimum. The paper [2] below also did an empirical study of neural bandit methods, so the findings in [2] may also be compared/combined with those in this paper to potentially get more insights. For example, [2] also found that neural bandits tend to work better when the objective function is complicated.
In fact, the connection between BNN-surrogate BO and neural bandits has also been discussed by the concurrent work of Kristiadi et al. (2023) in the context of linearized-Laplace approximation. The recent work of [3] has also shown the potential of using NTK in kernel regression, which may also be an empirical justification for the potential of BO with NTK based surrogate.

[1] Sample-Then-Optimize Batch Neural Thompson Sampling, 2022.
[2] Empirical Analysis of Representation Learning and Exploration in Neural Kernel Bandits, 2021.
[3] Kernel Regression with Infinite-Width Neural Networks on Millions of Examples, 2022.

**(2)** Another minor point which can make the paper easier to read is that when referring to the appendix, it'll make it easier for the reader if the specific subsection is referred to instead of just "Appendix D".

**Questions:**

- I find the contrast between deep kernel learning and deep ensemble particularly interesting, because both methods are able to use the strong representations learned by neural networks. I suppose the reason why deep kernel learning works better is because it can be readily plugged into BO which will take care of the exploration, while deep ensemble doesn't do well in exploration. Please see if this makes sense.

---

> ### Author Response · Authors · 2023-11-20
> **Author Response to Reviewer 6qj4**
>
> Thank you for your thoughtful and supportive comments! We really appreciate it.
>
> > Discussion of non-Bayesian neural networks as the surrogate in BO
>
> Thank you for the references! While our paper focuses on Bayesian neural network surrogates for Bayesian optimization, for a thorough investigation, we recognize that there are other works focusing on non-Bayesian methods such as neural bandits and NTK. We have updated our related works section with the suggested papers, and we have also updated Appendix D.3 to include a discussion of the performance of deep ensembles in other works.
>
> > Deep ensembles do not do well in exploration
>
> As you pointed out in your review, we noticed that deep ensembles often plateau at low objectives. We explore this phenomenon in Appendix D.3, and showed that smaller training sizes can often lead to deep ensembles having *less* uncertainty. In the low-data regime, we show that the loss landscape is relatively smooth, making it more difficult to find diverse solutions, corresponding to distinct basins of attraction, through only re-initializing optimization runs. The lack of model diversity in the ensemble leads to smaller predictive uncertainty estimates and worse exploration. This makes deep ensembles less suitable surrogate models when there is a minimal number of data points. Interestingly, this is in contrast to the success of deep ensembles in general applications. More training data (i.e., objective queries) helps deep ensembles again become competitive.
>
> > Refer to specific sections of the Appendix
>
> Thank you for the suggestion! We have updated our draft accordingly.

---

### Author Response · Authors · 2023-11-20
**General Reponse**

We would like to thank everyone for your thoughtful feedback. We appreciate your supportive comments and thoughtful suggestions. In this general comment, we highlight some of our key findings and also present new experiments inspired by your reviews. **Note we respond to each reviewer individually in separate posts.**

The choice of surrogate model in Bayesian optimization is a fundamental design decision that has been largely overlooked. We provide a comprehensive investigation, evaluating the roles of representation learning, non-stationarity, stochasticity, and more for BNNs in BO. We also provide benchmarks on a diverse set of synthetic and real-world objectives problems.

**Our headline findings include:**
1) I-BNNs---never previously considered for BO---are particularly compelling, especially on high-dimensional problems. This finding highlights the value of non-Euclidean similarity metrics.
2) There is a significant performance difference between approximate inference procedures for Bayesian optimization. Notably, we find deep ensembles, which perform very competitively in standard settings, perform relatively poorly in Bayesian optimization, and we explain why.
3) Full-network stochasticity does not confer major benefits, with DKL being relatively competitive. We also show that standard GP stochasticity may confer less of a benefit that previously understood, in contrast to conventional wisdom.
4) Standard GPs are still competitive on standard benchmarks, despite significant advances in Bayesian neural networks.
5) Objective functions in BO are sufficiently different that no single model dominates, suggesting the need to tailor the surrogate model to the objective (in contrast to vision and NLP, for example).

**We additionally want to emphasize three key contributions of our paper:**
1) This is by far the most comprehensive investigation of neural network-inspired surrogates for Bayesian optimization to date. The study also includes several novel choices of surrogates, notably infinite-width BNNs, which perform surprisingly well, and several novel conceptual findings outlined above. For calibration, compare for example to [Springenberg et al. (2016)](https://proceedings.neurips.cc/paper_files/paper/2016/file/a96d3afec184766bfeca7a9f989fc7e7-Paper.pdf), which was a NeurIPS oral (top 2% accepted papers) evaluating an SGHMC-BNN surrogate for BayesOpt. This is a highly successful paper, but it clearly does not contain nearly as many experiments or findings as our submission. While it is always possible to request more experiments or comparisons, our submission is exceptionally comprehensive relative to most published work on Bayesian optimization, which typically considers a smaller set of datasets and surrogate models.
2) The choice of surrogate for Bayesian optimization is fundamental and relatively underexplored, especially compared to other design decisions such as the choice of acquisition function. Our findings are surprising and have the potential to have a **significant** impact in championing new surrogate functions as a research area. We will release our experimental setup publicly so that it can be used to further benchmark and understand new surrogates.
3) While we do consider novel surrogates and settings, we have made a concerted effort not to have a horse in the race. There is no one single conclusion, but rather many nuanced findings, and **we deliberately avoid implying that BNNs are either better or worse than standard GPs for Bayesian Optimization**, but rather highlight their particular strengths and weaknesses in different settings. We hope the scientific value of this approach can be considered.

Inspired by reviewer feedback, we have put a substantial effort into providing several new experimental findings focusing on the optimization of GP and BNN hyperparameters, as detailed in Appendix D.12 in the updated manuscript. We provide empirical results showing the performance of GPs and BNNs with hyperparameters optimized on either a per-iteration or per-trial basis, and find our conclusions to be robust to the method of hyperparameter optimization. We have also updated Appendix D.11 with a new experiment addressing how the surrogate models perform under runtime constraints assuming fast function queries.

We hope reviewers can consider our response in their final assessment.

---

### Meta-Review · Area_Chair_jZ57 · 2023-12-08

**Metareview:**

This paper conducts a empirical investigation of representative variants of Bayesian neural networks as surrogate functions for Bayesian optimization. It gains several insights and provide discussion on the strength of different variants and their respective problem setting in comparison with standard GPs.

All reviewers consider the study of BNNs as a BO surrogate important and appreciate the good presentation of the paper. Two reviewers find the investigation comprehensive and provide useful insights for BO researchers and practitioners.

One reviewer had concerns about the number of datasets in empirical studies, the effect of function evaluation runtime in the comparison, and the effective impact of this paper. The authors' rebuttal have resolved most of them. The reviewer was still not fully convinced of the effective impact, but agree to raise his/her rating to acceptance.

**Justification For Why Not Higher Score:**

The topic is limited on a specific design choice of surrogate functions in BO using BNN variants. While the results provide useful insights to understand the strengths of different models, it is not clear how the findings could lead to a better algorithmic design.

**Justification For Why Not Lower Score:**

Consensus from all reviewers on the acceptance.

---

### Decision · Program_Chairs · 2024-01-16

Accept (poster)